# Immobility responses are induced by photoactivation of single glomerular species responsive to fox odour TMT

Harumi Saito[1], Hirofumi Nishizumi[1,2], Satoshi Suzuki[2], Hideyuki Matsumoto[3], Nao Ieki[3], Takaya Abe[4], Hiroshi Kiyonari[4,5], Masahiko Morita[6], Hideo Yokota[6], Nozomi Hirayama[7], Takahiro Yamazaki[1], Takefumi Kikusui[7], Kensaku Mori[3] & Hitoshi Sakano[1]

Fox odour 2,4,5-trimethyl thiazoline (TMT) is known to activate multiple glomeruli in the mouse olfactory bulb (OB) and elicits strong fear responses. In this study, we screened TMT-reactive odourant receptors and identified *Olfr1019* with high ligand reactivity and selectivity, whose glomeruli are located in the posterodorsal OB. In the channelrhodopsin knock-in mice for *Olfr1019*, TMT-responsive olfactory-cortical regions were activated by photostimulation, leading to the induction of immobility, but not aversive behaviour. Distribution of photo-activation signals was overlapped with that of TMT-induced signals, but restricted to the narrower regions. In the knockout mice, immobility responses were reduced, but not entirely abolished likely due to the compensatory function of other TMT-responsive glomeruli. Our results demonstrate that the activation of a single glomerular species in the posterodorsal OB is sufficient to elicit immobility responses and that TMT-induced fear may be separated into at least two different components of immobility and aversion.

[1] Department of Brain Function, Faculty of Medical Sciences, University of Fukui, Matsuoka, Fukui 910-1193, Japan. [2] Department of Biological Sciences, Graduate School of Science, The University of Tokyo, Bunkyo-ku, Tokyo 113-0032, Japan. [3] Department of Physiology, Cellular and Molecular Physiology, Graduate School of Medicine, The University of Tokyo, Bunkyo-ku, Tokyo 113-0033, Japan. [4] Genetic Engineering Team, RIKEN, Center for Life Science Technologies, Kobe, Hyogo 650-0047, Japan. [5] Animal Resource Development Unit, RIKEN, Center for Life Science Technologies, Kobe, Hyogo 650-0047, Japan. [6] Image Processing Research Team, RIKEN, Wako, Saitama 351-0198, Japan. [7] Department of Animal Science and Biotechnology, School of Veterinary Medicine, Azabu University, Sagamihara, Kanagawa 252-5201, Japan. Correspondence and requests for materials should be addressed to Hi.S. (email: sakano.hts@gmail.com).

In the mammalian brain, sensory information is spatially encoded and forms neural maps that are fundamental for higher-order processing. In the mouse olfactory system, odour signals detected by olfactory sensory neurons (OSNs) in the olfactory epithelium (OE) are represented as an odour map in the olfactory bulb (OB)[1–3]. Because each glomerulus corresponds to a single odourant receptor (OR)[3,4], and a single odourant can interact with multiple OR species[5,6], odour signals received in the OE are converted into a topographic map of multiple glomeruli activated in varying magnitudes[2,5–9]. The main olfactory system in rodents is known to mediate innate aversive behaviours to spoiled foods, and fear responses to predator odours[10–14]. One predator odourant is 2,4,5-trimethyl thiazoline (TMT), a molecule secreted from the fox anal gland. TMT induces strong fear responses in rodents including not only aversion but also immobility that is not induced by other aversive odours[11,12]. We previously reported mutant mice, ΔD[10], in which glomeruli were ablated by the zone-specific expression of diphtheria toxin in the dorsal OB, and demonstrated their failure to elicit innate fear upon exposure to TMT. Optical imaging of the dorsal OB has revealed that these TMT-responsive glomeruli in the dorsal domain are clustered to a posterior part of the $D_{II}$ subdomain[9]. Although multiple glomeruli are activated in both dorsal and ventral domains by TMT, it was uncertain whether individual glomeruli are functionally specialized to instruct particular responses, or whether a pattern of activated glomeruli as a whole is recognized to mediate behavioural decisions. It was also unclear whether the TMT-induced fear responses can be further divided into different categories, such as immobility and avoidance. To address these questions, we performed loss-of-function and gain-of-function experiments to determine the effect on mouse behaviour when a single glomerular species, responsive to the fox odourant TMT, is deleted or photoactivated.

## Results

**Isolation of cDNAs for TMT-responsive ORs.** For gene-targeting experiments, we searched for candidate ORs with high affinity and selectivity for TMT. Although TMT activates multiple glomeruli in both the dorsal and ventral OBs[10], the glomeruli responsible for inducing innate fear appear to be confined to the $D_{II}$ subdomain of the OB[9]. We, therefore, analysed the dorsal OB by optical imaging, after exposure to varying concentrations of TMT (Fig. 1a and Supplementary Fig. 1a). A related odourant 4-methyl-thiazoline (4MT) that induces aversion but not immobility[14] was examined in parallel. A large area of the dorsal OB was activated in the presence of high concentrations of TMT. However, the number of activated glomeruli narrowed as the TMT concentration was lowered. These glomeruli were located in the posterior part of the dorsal OB (Fig. 1a). To clone the OR genes associated with these glomeruli, we retrogradely labelled the connecting OSNs[15,16] by injecting DiI solution into the single glomeruli activated by TMT, but not by 4MT. At 36 h after the DiI injection, we confirmed that the dye had specifically stained OSNs in the dorsal OE (Supplementary Fig. 1b). OE cells were dissociated and DiI-positive OSNs were subjected to single-cell PCR with reverse transcription for OR gene cloning[5,15]. After the completion of five independent experiments, 12 OR complementary DNAs (cDNAs) frequently and repeatedly isolated were used for further analyses (Supplementary Table 1a). Among them, we selected *Olfr1019*, *Olfr57* and *Olfr30* based on their high affinities for TMT measured by the cyclic AMP (cAMP)-induced luciferase assay[6,17] (Supplementary Fig. 1c). Confirmation that these OR genes are indeed expressed in the dorsal OE was obtained by *in situ* hybridization (Supplementary Fig. 1f). Independent of this DiI screening, we

analysed 266 previously reported ORs[18] for their interactions to TMT by high-throughput screening in which TMT-responsive ORs were analysed not only in a posterior part of the $D_{II}$ subdomain but also in other areas of the OB (Supplementary Fig. 1d and Supplementary Table 1b). Combined with the three candidate ORs identified by the DiI experiment, a total of 18 ORs were further analysed for their dose responsiveness to TMT (Supplementary Fig. 1b). Six ORs demonstrating high TMT responsiveness were then subjected to ligand selectivity tests using 17 different odourants (Fig. 1c and Supplementary Fig. 1e). Among them, *Olfr1019* showed the greatest specificity to TMT and was used for further gene-targeting experiments.

**Knock-in and knockout mice for *Olfr1019*.** For the loss-of-function and gain-of function experiments of the Olfr1019 glomeruli, we generated mutant mice in which the *Olfr1019* gene was either deleted or linked to the coding sequence of channelrhodopsin (ChR) (Fig. 2a, and Supplementary Fig. 2a,b). For the knock-in (KI) experiment, ChR wide receiver (ChRWR) was used in place of conventional ChR2. ChRWR is a chimeric molecule of ChR2 and ChR1 and possesses advantages over ChR2, such as improved expression in the plasma membrane and enhanced photocurrent with smaller desensitization[19]. In the KI, ChRWR was fused with a fluorescent marker, Venus, so that OSN axons expressing the KI allele of *Olfr1019* could be visualized. Venus-positive OSNs were all confined to the dorsal OE, and no Venus signal was found in the vomeronasal organ. As expected, a pair of fluoresced glomeruli was detected for Olfr1019 in the posterior $D_{II}$ subdomain of the OB (Fig. 2b,c and Supplementary Fig. 2c,d). Although the locations of Olfr1019-KI glomeruli varied among animals, they were primarily located within the posterior $D_{II}$ subdomain as identified by optical imaging upon exposure to TMT (Supplementary Fig. 2c). The KI glomeruli were rarely found in the lateral aspect of the dorsal OB where the 4MT-responsive glomeruli were clustered. We also analysed the knockout (KO) mice for the expression of the deleted *Olfr1019* allele (Fig. 2c,d). Extended yellow fluorescent protein (EYFP)-positive OSN axons expressing the KO allele did not converge to a specific glomerulus, but were dispersed across the dorsal OB (Fig. 2d). When the OB sections were analysed by immunohistochemistry, OSN axons expressing the *Olfr1019* KO allele were scattered and detected within various D-domain glomeruli. *In situ* hybridization revealed that OSNs expressing the *Olfr1019* KO allele had activated another OR gene member to maintain the one neuron-one receptor rule[4]. These coexpressing secondarily chosen OR genes were not necessarily from the *Olfr1019* cluster or from our gene list for TMT-responsive ORs (Fig. 2d).

**Photoinduced spike-discharges in M/T cells.** Mitral/tufted (M/T) cells are secondary neurons receiving odour signals from connecting glomeruli (Supplementary Fig. 3a) and send odour information to various regions in the olfactory cortex (OC)[20–25]. To determine whether stimulation of a single glomerular species could induce spike discharges in M/T cells, the KI mice were photoilluminated and action potentials of M/T cells were analysed by single-unit recording (Fig. 3a–c and Supplementary Fig. 3c). Layer locations of the tip of the recording microelectrode in the OB were examined by monitoring the configuration of lateral olfactory tract (LOT)-evoked field potentials (Supplementary Fig. 3a,b)[9,24,25]. We first studied the responsiveness of cells in the KI-OB to different dose intensities of photo exposure (Fig. 3b). In the external plexiform layer (EPL) unit, spike responses were saturated at 0.1 mW mm$^{-2}$, whereas in the mitral cell layer (MCL) unit, responses were gradually increased and saturated at 1.0 mW mm$^{-2}$. Consistent with a previous study demonstrating that tufted cells are more sensitive

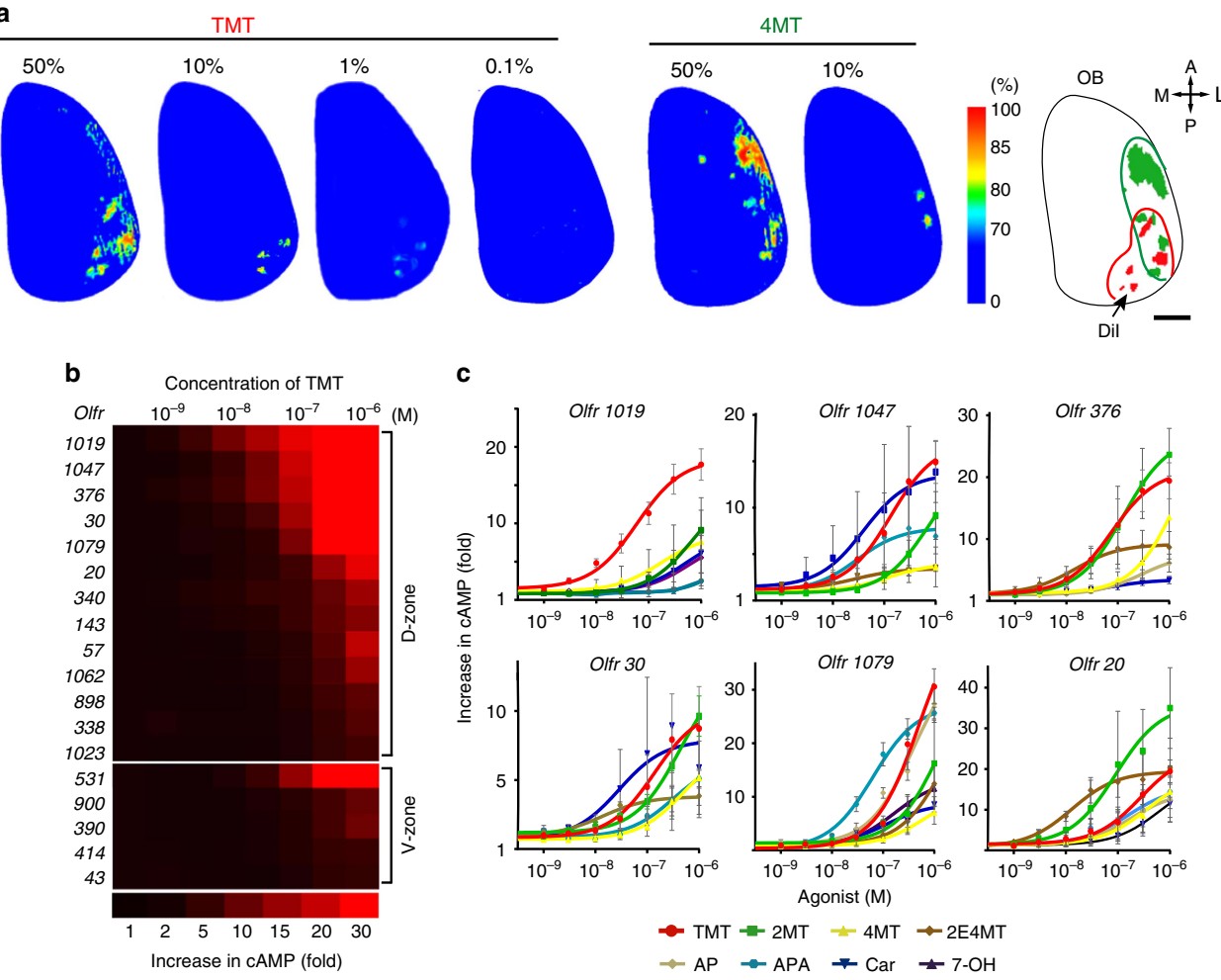

**Figure 1 | Isolation of TMT-responsive *OR* genes.** (**a**) Optical imaging in the olfactory bulb (OB). Dorsal views of the right OB are shown. To identify glomeruli that specifically responded to the fox odourant, TMT, glomerular activity was analysed by optical imaging. Activities are shown in rainbow colours. Diluted TMT solutions (0.1, 1, 10 and 50%) were presented to mice. A related odourant, 4MT (10 and 50%), that induces aversive responses was used to exclude any 4MT-responsive glomeruli that crossreact with TMT. Activation areas for TMT (red) and 4MT (green) in the dorsal OB are schematically shown on the right. A, anterior; P, posterior; M, medial; L, lateral. Scale bar, 500 μm. (**b**) Dose responsiveness of ORs to TMT. Eighteen *OR* gene clones selected from the two screened sets were individually introduced into HEK cells to analyse the expressed OR molecules for their interactions to TMT. (**c**) Six different ORs that demonstrated high affinity to TMT were analysed by the luciferase assay for their interactions to eight different agonists at various concentrations. All values are in mean ± s.e.m.; $n = 4$. TMT, 2,4,5-trimethyl thiazoline; 2MT, 2-methyl-thiazoline; 4MT, 4-methylthiazoline; 2E4MT, 2-ethyl-4 methyl-thiazoline; AP, acetophenone; APA, allyl-phenyl-acetate; Car, carvone; 7-OH, heptanol.

than mitral cells to lower concentrations of the TMT[20], spike responses of the EPL unit were more sensitive to weaker light than the MCL unit (Fig. 3b). We then examined response profiles of M/T cells to different durations of stimulation with the light intensity of 1.3 mW mm$^{-2}$ (Fig. 3c and Supplementary Fig. 3c). The EPL unit generated initial burst discharges of over 100 Hz immediately after the onset of 20 ms or longer durations of illumination (Fig. 3c and Supplementary Fig. 3c). In contrast, the MCL unit demonstrated spike responses to 50 ms or longer illumination (Fig. 3c). When durations of illumination were sufficient to induce spike responses, the photoilluminated cells yielded burst discharges at 50 Hz with delayed onset (Fig. 3c and Supplementary Fig. 3c). These results indicate that over 200 ms of photoillumination induces the plateau of spike rates in both the EPL and MCL units (Fig. 3a,c and Supplementary Fig. 3c).

**TMT-responsive OC regions are partly activated by Olfr1019.** To examine whether TMT-responsive brain regions are also activated by photostimulation of Olfr1019 glomeruli in the KI mouse,

we studied the expression of Egr1, an immediate-early protein induced by neuronal activity, in the OB and in the following brain regions: anterior olfactory nucleus (AON), anterior piriform cortex, olfactory tubercle (OT), cortical amygdala (CoA), medial amygdala (MeA) and bed nucleus of stria terminals (BNST) (Fig. 4a–c and Supplementary Fig. 4a–h). To set up the condition of photoillumination, we examined Egr1 expression with three different illumination conditions: 2 Hz light in 20 ms pulses at 1.3 mW mm$^{-2}$ for 30 min, 2 Hz light in 250 ms pulses at 1.3 mW mm$^{-2}$ for 15 min, and 2 Hz light in 250 ms pulses at 1.3 mW mm$^{-2}$ for 30 min. With the 20 ms stimulation, Egr1 signals were faint and difficult to detect (Supplementary Fig. 4b–h). In the experiment using the illumination condition of 250 ms pulses with 250 ms interval for 15 min, the result was basically the same as that with the stimulation for 30 min (Supplementary Fig. 4b–f). However, the signals were much weaker in the MeA and BNST (Supplementary Fig. 4g,h). We, therefore, chose to use the illumination condition of 250 ms pulses with 250 ms interval for 30 min. As a positive control, TMT (1 and 10%) absorbed onto a filter paper was presented to the wild-type (WT) mice (Fig. 4a–c

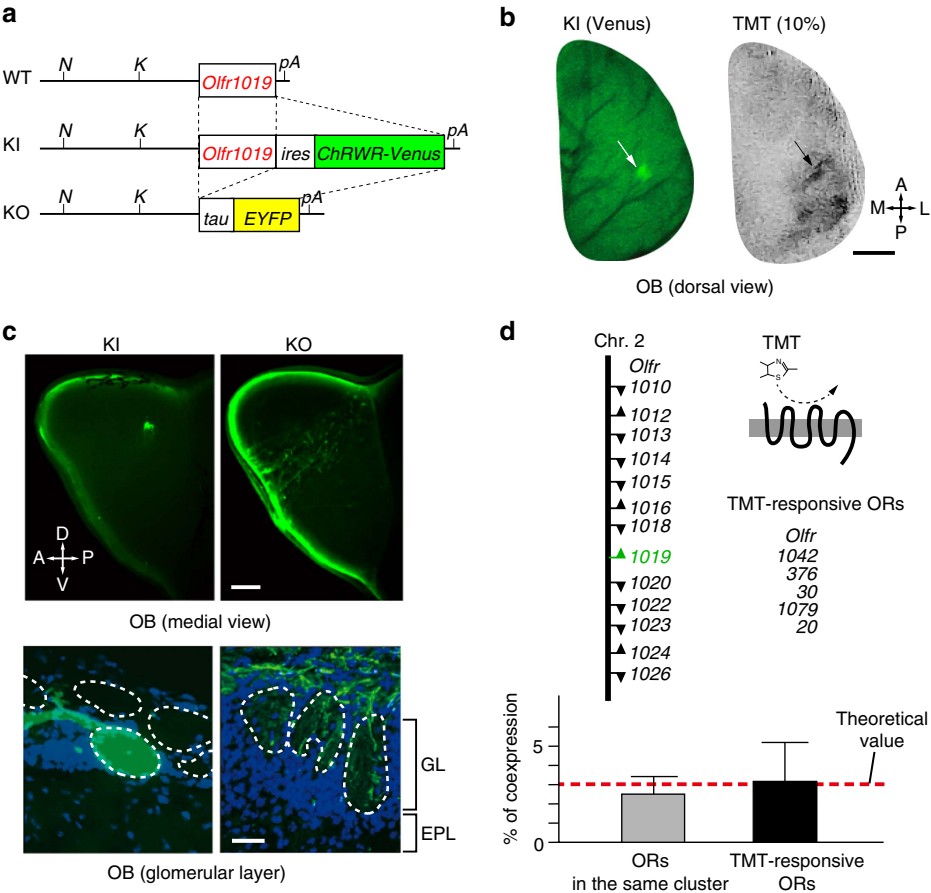

**Figure 2 | Knock-in and knockout mice for *Olfr1019*.** (**a**) Genetic structures of the *Olfr1019* loci for wild-type (WT), knock-in (KI) and knockout (KO) mice. In KI mice, genes for a channelrhodopsin wide receiver (ChRWR) and a fluorescent marker (Venus) were linked with the *ires* sequence to the coding region of *Olfr1019*. In KO mice, the deleted allele of *Olfr1019* was labelled with the *tau-EYFP*. Restriction-enzyme cleavage sites (*N*, *Nhe*I; *K*, *Kpn*I) and polyA sites (*pA*) are shown. (**b**) Dorsal views of the KI mouse OB. A fluorescent glomerulus was detected for Olfr1019 in the dorsal OB of KI mice (left). The KI mouse OB was also analysed by optical imaging after exposure to 10% TMT (right). Scale bar, 500 μm. A, anterior; P, posterior; M, medial; L, lateral. (**c**) OB of KI and KO mice. Targeting of OSN axons expressing the KI and KO alleles of *Olfr1019* are compared on the medial surface of the OB (top). Scale bar, 200 μm. OB sections of KI and KO mice were also analysed by immunohistochemistry with anti-GFP antibody (bottom). In KI-OB, a discrete glomerular structure was formed for *Olfr1019*. In KO mice, fluorescence-labelled axons (green) were dispersed and targeted into various glomeruli in the dorsal OB. Scale bar, 10 μm. GL, glomerular layer; EPL, external plexiform layer; A, anterior; P, posterior; D, dorsal; V, ventral. (**d**) Double *in situ* hybridization of the OE. To examine coexpression of the KO allele of *Olfr1019* with other OR genes, two sets of mixed probes were used. One set included 12 different *OR* genes that belong to the same *Olfr1019* cluster.: *Olfr1010, 1012–1016, 1018, 1020, 1022–1024* and *1026*. The other set was a mixed probe of five *OR* genes coding for TMT-responsive ORs. All probes included entire coding sequences. Transcriptional directions are indicated by arrowheads. Coexpression rates (%) are shown in the bottom. All values are in mean ± s.e.m.

and Supplementary Fig. 4b–h). We also analysed TMT-exposed Olfr1019-KO mice, vanillin-exposed WT mice and photo-illuminated M72-ChR2-YFP mice[26] for comparisons (Supplementary Fig. 4c–h). In the photoilluminated KI mice, Egr1 signals were found in periglomerular cells surrounding the Olfr1019 glomerulus and in the M/T cells right underneath (Supplementary Fig. 4b). In contrast, in the TMT-exposed WT mice, activated cells were detected in the broader region surrounding the Olfr1019 glomerulus (Supplementary Fig. 4b).

To identify the OC regions specifically activated in the photoilluminated KI mice, we compared the distribution patterns of Egr1 in the KI mice with those in the Olfr1019-KO and M72-Ch2-YFP mice (Fig. 4a–c and Supplementary Fig. 4c–h). Egr1 signals were found to be prominent in some cortical regions of the photoilluminated KI mice: in the AON, signals were notably high in the posterodorsal region (Fig. 4a, Supplementary Fig. 4c and Supplementary Movie 1); in the lateral OT, the anterior part of cap compartments was intensively labelled among three different components, cortical zone, clusters of interneurons in

cap compartment and islands of Calleja (Fig. 4b, Supplementary Fig. 4e and Supplementary Movie 2). Previous tracing experiments revealed topographic correlations between the CoA and dorsal-region glomeruli that appear to be important in inducing innate aversive responses[14,21]. In the CoA and MeA, Egr1 signals were high in the anterior parts (Fig. 4c, and Supplementary Fig. 4f,g). In contrast, the signals were much reduced in these areas in the TMT-exposed KO mice (Fig. 4a–c and Supplementary Fig. 4c–g). The amygdalopiriform transition area (AmPir) was reported to be responsible for inducing stress responses[27]. Egr1 expression was not enhanced in the AmPir in the photoilluminated KI, although 10% TMT fully activated the AmPir in the WT (Supplementary Fig. 4f). In the BNST, Egr1 signals were high in the anterodorsal to medial regions in the photoilluminated KI mice. However, signals were high in the posterior region in the TMT-exposed WT mice (Supplementary Fig. 4h). In the TMT-exposed KO, Egr1 signals were reduced but still remained in the fear-responsive OC regions and in the BNST[28] (Supplementary Fig. 4c–h). It is likely that in the KO,

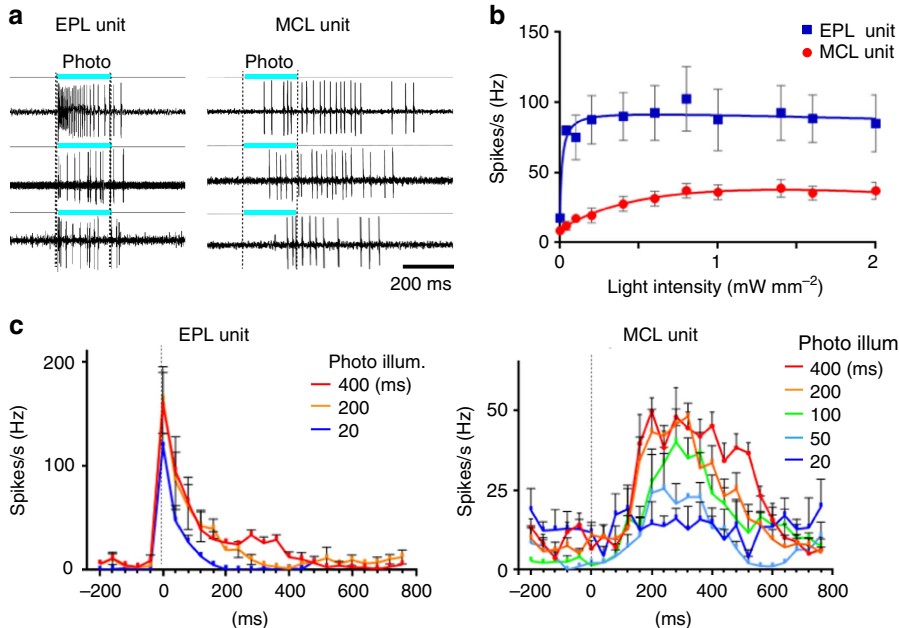

**Figure 3 | Spike discharges induced in *Olfr1019* glomeruli in the photoilluminated KI mouse.** (**a**) Light-evoked spike discharges in the external plexiform layer (EPL) and mitral cell layer (MCL) units. Three examples are shown for EPL and MCL units, respectively. Durations of the photoillumination (200 ms) is indicated by cyan bars. (**b**) Extracellular single-cell recording with various light intensities in the KI-OB. Dose–response curves show the spike discharge rate in the different light intensity with the range from 0.04 to 2.0 mW mm$^{-2}$. EPL unit (blue), $n = 4/45$ tested; MCL unit (red), $n = 4/60$ trails. (**c**) Spike-rate analysis in the EPL and MCL units. Various durations of light stimulation were given from 20 s. All values are in mean ± s.e.m. For ECL units, $n = 5/55$ trials for 20 ms; $n = 7/65$ trials for 200 ms; $n = 6/60$ trials for 400 ms. For MCL units, $n = 4/45$ trials for 20 ms; $n = 3/35$ trials for 50 ms; $n = 5/55$ trials for 100 ms; $n = 7/80$ trials for 200 ms; $n = 5/60$ trials for 400 ms.

other TMT-responsive glomeruli remaining in the posterior $D_{II}$ subdomain had activated the fear circuit to compensate for the function of deleted Olfr1019 glomeruli.

**Photoactivation of Olfr1019 glomeruli induces immobility**. As photostimulation of Olfr1019-ChRWR glomeruli was able to activate TMT-responsive brain regions, we next examined whether fear responses could be induced in the photoilluminated KI mice. WT and KI mice were analysed in parallel for various types of behaviour (Fig. 5a,b,d and Supplementary Fig. 5b,d). The KI mouse whose head had been shielded with an aluminum foil was also analysed as a negative control. Mice (7–12 weeks old) were kept without light for the first 3 min of experiment, and then given optical stimulation. We first examined immobility of KI mice with the different pulse conditions (20, 50, 100 and 250 ms pulses at 1.3 mW mm$^{-2}$) and intensities (250 ms pulse at 0.3 and 1.3 mW mm$^{-2}$) of photoillumination (Supplementary Fig. 5a,e). Our results indicated that 250 ms pulse at 1.3 mW mm$^{-2}$ intensity induced immobility more effectively than other pulse conditions. We then performed two sets of experiments with different durations of illumination. In one set, 5 s pulse was given every minute (Fig. 5a–d, Supplementary Fig. 5e and Supplementary Movie 3), whereas in the second set, 1 s pulse was given every 20 s (Supplementary Fig. 5b,d and Supplementary Movie 4). As shown in the video (Supplementary Movies 3 and 4), the KI mice demonstrated prolonged immobility responses by holding whisker movement and crouching the body down. In contrast, control mice (WT and shielded KI) continued to demonstrate various natural behaviours, for example, walking, stopping, exploring and grooming, despite photoillumination (Fig. 5a–c, Supplementary Fig. 5b,c and Supplementary Movies 3 and 4). To further exclude the possibility that photoactivation of any single glomerulus, regardless of the ligand specificity, could induce immobility responses, we analysed M72-ChR2-YFP

mice[26] in which the *ChR2* gene was introduced into the *M72 OR*-gene locus. It has been shown that photostimulation of this mice generates sufficient signals in M72 glomeruli that can be transmitted to the central brain for memory-based go/non-go task[26]. As M72 glomeruli are localized to the dorsal surface of the OB, we photoilluminated and analysed M72 mice in the same manner as the KI mice of Olfr1019-ChRWR. We found that M72 mice did not show any immobility responses to photoillumination and behaved the same as the WT control (Supplementary Fig. 5e).

We then examined whether the photostimulation of Olfr1019 glomeruli in the KI mice of Olfr1019-ChRWR could cause any stress reactions. After photoillumination, adrenocorticotropic hormone (ACTH) levels were not so much increased in the KI as in the TMT-exposed WT mice (Supplementary Fig. 5g), indicating that photoillumination did not induce stress responses in the KI. This observation is consistent with the low Egr1 expression in the AmPir in the photo-illuminated KI mice but high in the TMT-exposed WT (Supplementary Fig. 4f). It is interesting, in this connection, that KI mice became immobile, but did not exhibit obvious avoidance behaviours upon photoillumination (Fig. 5d and Supplementary Fig. 5d). We also analysed Olfr1019-KO mice for TMT responsiveness (Fig. 5e,f, and Supplementary Fig. 5f). When subjected to 10% TMT, the WT demonstrated clear fear responses including immobility. However, the KO mice behaved differently upon stimulation with TMT, demonstrating lower immobility than that of the WT mice (Fig. 5e and Supplementary Fig. 5f). Interestingly, immobility responses towards TMT were not entirely abolished in the KO mice. This is likely due to the compensatory function of the remaining glomeruli in the posterior $D_{II}$ subdomain that are also TMT responsive. It should be noted that in this KO, aversive responses to TMT were not significantly affected (Fig. 5f).

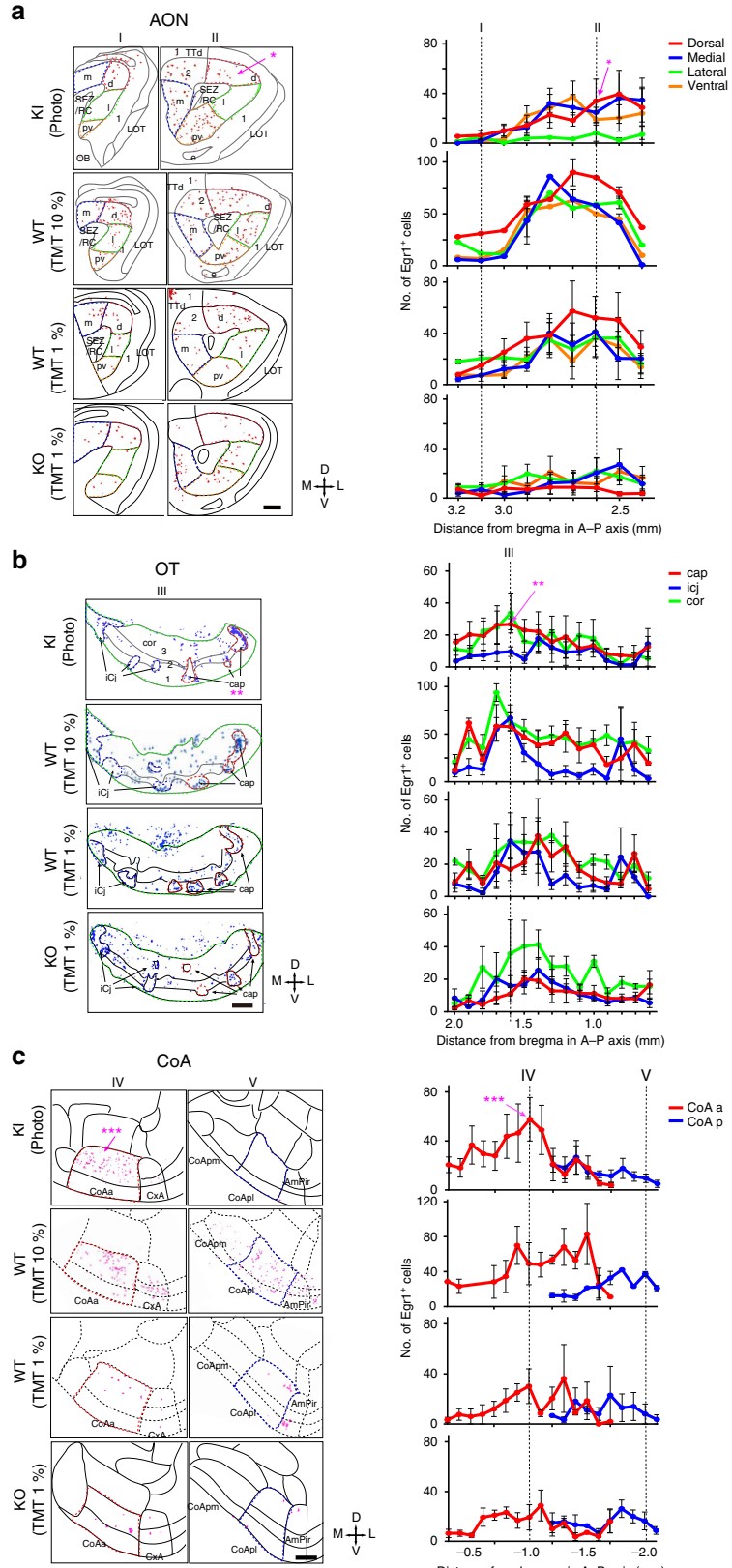

## Discussion

In this study, we screened TMT-responsive ORs and identified Olfr1019 with high ligand affinity and selectivity. Our present data collectively demonstrate that stimulation of a single glomerular species for Olfr1019 is sufficient to elicit innate immobility. TMT usually activates multiple glomeruli and induces both avoidance and immobility responses in rodents[10,11,12,14]. In our optogenetic experiments, however, photoactivation of a single glomerular species for Olfr1019 induced immobility, but not clear aversion, indicating that this OR is specialized for only an immobility response. In this connection, Egr1 experiments demonstrated that photoactivated areas in the AON, OT, CoA, MeA and BNST were restricted to smaller areas compared with those activated by 10% TMT. Although the posterior regions of CoA are reported to be responsible for TMT-induced aversive responses, and AmPir is reported to be responsible for TMT-induced stress hormone responses[14,27], these regions are less activated in the photoilluminated KI mice. These results are consistent with our behavioural analyses that the photoilluminated KI mice did not demonstrate aversive responses. It is likely that TMT-induced fear responses can be divided into at least two distinct components, immobility and avoidance, whose functional domains in the glomerular map may be separate. Because TMT activates not only the dorsal domain of immobility but also in other OB areas including the ventral region[9,10], these nondorsal glomeruli may also contribute to the activation of OC, MeA and BNST.

Based on our previous studies[2,8–10], we propose that the olfactory map is not merely a projection screen for pattern recognition of activated glomeruli but is also composed of functional subdomains for innate odour responses. As seen in the fly[29], olfactory circuits in mice appear to be hard-wired for their innate responses. However, during evolution, the olfactory system could freely move out or move in a particular glomerulus from or to a specific functional domain in the OB by changing the expression levels of axon guidance molecules[30,31]. In this scenario, animals could easily adapt their olfactory system during evolution to the changes in the odour environment, such as appearance of new predators. It is generally thought that an activation pattern of a specific set of glomeruli is transmitted to the OC to discriminate and identify a particular odourant[2,20–23]. However, the olfactory system uses a different strategy to elicit innate fear responses than that of odour discrimination. Our present study clearly demonstrates that photoactivation of a single glomerular species for Olfr1019 is capable of inducing immobility responses in the ChRWR-KI mice, although multiple glomeruli are activated by TMT in both the dorsal and ventral regions. It appears that innate immobility could be induced once the M/T cells within the fear domain in the posterior $D_{II}$ region were activated even by a single glomerular species. Our KI mice will serve as an excellent model system to precisely identify the neural circuit from single glomeruli to specific areas in the brain regions.

## Methods

**Optical imaging and DiI staining.** For optical imaging, odourants were presented to mice in a glass test tube after diluting them in mineral oil. For each recording trial, data were collected for 8 s with a frame length of 0.5 s (16 frames per trial). Each odourant was tested at least three times per animal. After imaging, 3% DiI in dimethylformamide was pressure-injected into the OB at the centre of an odour-activated glomerulus[8,15,16]. Odourants were obtained from Contech Inc., Sigma-Aldrich and Wako Pure Chemical Industries Ltd.

**OR-gene cloning.** At 36 h after DiI injection, mice were anaesthetized, the OE was dissected and cells were dissociated with dispase (50 µg ml$^{-1}$, Invitrogen) and collagenase (1 g ml$^{-1}$, Invitrogen) at 37 °C for 10 min. DiI-stained cells were collected using a micromanipulator under a fluorescence microscope and used for single-cell PCR with reverse transcription[5,15]. To avoid contaminations, all samples were reexamined by PCR cloning. Primer sequences for PCR were 5′-ATGGCCTACGACCGGTACGTNGCNATHTG-3′ and 5′-TGGGAGGCGC AGGTGSWRAANGCYTT-3′.

**Luciferase assay.** Dual-Glo Luciferase Assay System (Promega) was used in our study[6,17]. We used CRE-luciferase (Stratagene) to measure an increase in the intracellular cAMP; *Renilla* luciferase driven by a constitutively active SV40 promoter (pRL-SV40; Promega) served as an internal control for cell viability and transfection efficiency. Hana3A cells were plated on poly-D-lysine-coated 96-well plates (BioCoat; Becton Dickinson). We transfected expression vectors for ORs with Rho tag in a Hana3A cell line along with vectors for RTP1S, CRE-luciferase and pRL-SV40 using Lipofectamine2000 (Invitrogen). Transfected cells were incubated with odourants for 12–16 h. The data were analysed using GraphPad Prism 4.

**Animal preparation.** Gene targeting for the *Olfr1019* was performed following the standard procedure[32]. Briefly, DNA fragments containing2 *Olfr1019* were isolated from C57BL/6 BAC clones (RP23-384L16 BACPAC) and used for targeting. To generate KI (Accession No. CDB1182K) and KO mice (Accession No. CDB1008K: http://www2.clst.riken.jp/arg/mutant%20mice%20list.html) and KO mice, ires-ChRWR-Venus-loxP-neo-loxP and Tau-YFP-loxP-neo-loxP were inserted into the BAC clones just downstream and upstream of the *Olfr1019* sequence, respectively. A 13 kb of DNA fragment with 5.2 kb of 5′- and 3 kb of 5′-arms was subcloned into the pBAD vector. The RED-ET recombination kit (Gene Bridges) was used for inserting the gene cassette and subcloning the targeting vector. The neomycin selection cassette was obtained from Gene Bridges (A003). The ChRWR plasmid of a C-terminal fusion construct with Venus was made as previously described[19]. M72YFP-ChR mice were purchased from the stock (021206) in the Jackson Lab[26]. Animal experiments were performed in accordance with the guideline of Animal Care Committees in the University of Tokyo, University of Fukui, Azabu University and RIKEN Kobe Branch.

**Electrophysiological recording.** Sixteen male mice (7–12 weeks old) were used for the single-unit recordings. For extracellular single-unit recordings from M/T cells, ~1 mm diameter of the skull above the ChRWR-Venus-positive glomeruli was removed using fine forceps. A glass micropipette (50–100 MΩ) filled with 4 M NaCl was set right above or adjacent to the Venus-positive glomeruli[20]. Single-unit discharges were recorded from the EPL or MCL. Signals of single-unit discharges were amplified (MEZ-8301, Nihon Kohden, Tokyo, Japan), filtered (150-3 kHz; AB-610J, Nihon Kohden) and stored in a computer via an AD converter with the Spike2 software (Cambridge Electronic Design, Cambridge, UK). We identified one photoactivated M/T cell after 10–60 trials. We performed at most ~20 trials for each glomerulus due to the damage caused by the electrode insertion.

The LOT-evoked field-potentials were recorded in the OB to determine the layer specificity of the cells[24,25]. A concentric electrode was inserted into the LOT (2.0 mm anterior to the bregma, 2.3 mm lateral from the midline and ~5.5 mm from the brain surface) for electrical stimulation. Square current pulses (100 µs duration, 20–100 µA) were used to electrically stimulate the LOT[24,25]. Mitral cells were identified by their antidromic spike responses to the LOT stimulation.

**Figure 4 | Examination of the neural activity in the OC by Egr1 expression.** Brain sections were analysed by immunohistochemistry for the activity-dependent expression of Egr1, an immediate-early gene product. KI mice of Olfr1019-ChRWR were photoilluminated with a 2 Hz light of 250 ms pulses for 30 min (KI, WT and M72-ChR2-YFP mice). WT and KO mice were exposed to 1 and 10% TMT for 30 min. Egr1 signals were distinguished from background by binarizing the images. The following OC regions were analysed for Egr1 expression: (**a**) AON, anterior olfactory nucleus; (**b**) OT, olfactory tubercle; and (**c**) CoA, cortical amygdala. LOT, lateral olfactory tract; m, medial part of AON; d, dorsal part of AON; l, lateral part of AON; pv, posteroventral part; SEZ/RC, subependymal zone; TTd, Taenia tecta dorsal part; 1,2,3, layers 1, 2 and 3; cor, cortical zone; icj, islands of Calleja; CoA a, pl, pm, anterior, lateral posterior and medial posterior cortical amygdala area; CxA, cortex–amygdala transition zone; AmPir, amygdalopiriform transition area. The *, ** and *** are the OC regions (AON, OT and CoA) where the Egr1 signals are prominent in the photoilluminated KI. Quantifications of Egr1-positive neurons are shown in the right for the AON, OT and CoA (n = 5). The numbers of Egr1-positive neurons in various regions in OC sections are plotted along the anterior–posterior (A-P) axis (mm distance from the bregma). Positions of slices (I to V) are designated. Scale bars, 250 µm.

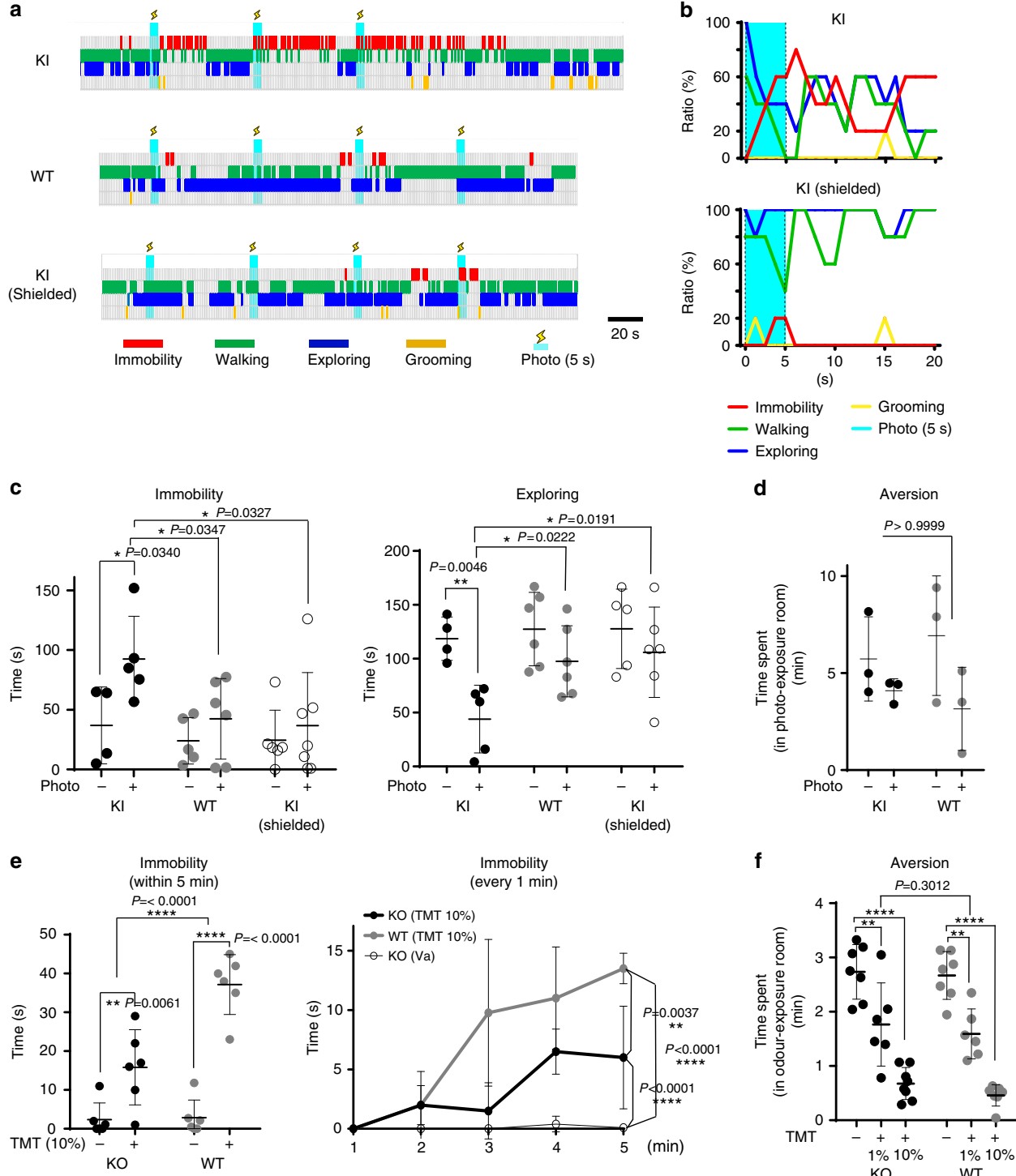

**Figure 5 | Behavioural analyses of KI and KO mice.** (**a**) Time-course studies of mouse behaviour after photoillumination. KI, WT and shielded KI mice were analysed. Immobility (red), walking (green), exploring (dark blue) and grooming (yellow) are shown. Timing of the photoillumination (2 Hz of 250 ms pulses at 1.3 mW mm$^{-2}$ for 5 s) is indicated (cyan bars with lightning marks). (**b**) Distribution of phototriggered behaviours in the KI and shielded KI mice. Photoillumination of 2 Hz was given for 5 s (cyan) with 250 ms pulses. Percentages of individual behaviours are shown; $n = 5$. (**c**) Quantification of immobility and exploratory behaviours. Immobility (left) and exploratory (right) behaviours were compared within 3 min after illumination among KI, WT and shielded KI mice. Asterisks indicate two-way analysis of variance (ANOVA) followed by Bonferroni correction. *$P < 0.05$, **$P < 0.01$. Error bars are ± s.d., $n = 5$ for KI and $n = 6$ for WT and shielded KI. (**d**) Open-field aversion analyses. KI and WT mice were photoilluminated when they stayed in one particular room out of two in a cage. Time lengths that each animal spent in the photo-exposure room are compared within 10 min ($n = 3$). $P$ values of two-way ANOVA followed by Bonferroni correction indicated in red. (**e**) Immobility responses to TMT are compared between Olfr1019-KO and WT mice. Duration of immobility was measured within 5 min (left) ($n = 6$) or every 1 min (right) ($n = 4$) after TMT presentation. Asterisks indicate two-way ANOVA followed by Bonferroni correction. **$P < 0.01$, ****$P < 0.0001$. Error bars are ± s.d. (**f**) Open-field aversion to TMT. A piece of filter paper immersed with 1 and 10% TMT was placed in one particular room out of two in a cage. The time spent in the odour-exposure room was measured for the WT and KO mice within 5 min. Asterisks indicate two-way ANOVA followed by Bonferroni correction. **$P < 0.01$, ****$P < 0.0001$. Error bars are ± s.d., $n = 6$ (1% TMT) and 8 (10% TMT).

**Photo- and odour-induced Egr1 expression in the OB and OC.** For the Egr1 histology experiments, 16 male mice (7–12 weeks old) were analysed. The photoillumination conditions are 20 ms pulse with 250 ms interval for 30 min, 250 ms pulses with 250 ms interval for 15 min and 250 ms pulses with 250 ms interval for 30 min. After photostimulation at the Olfr1019-KI glomeruli, animals were gently left for 60 min and transcardially perfused with saline followed by a fixative containing 4% paraformaldehyde. For odour exposure, mice were exposed to pieces of filter paper immersed with 1% TMT, 10% TMT or 10% Vanillin for 30 min, gently left for 60 min and perfused with 4% paraformaldehyde. Brains sections were prepared using a cryostat (HM500-M; Carl Zeiss) and immunostained by anti-Egr1 antibodies.

**Immunohistochemistry and _in situ_ hybridization.** Anti-Egr1 and anti-GFP antibodies were purchased from Cell Signaling Technology (cat no. mAb4153) and Abcam (cat. no. ab13970), respectively. The secondary antibodies, Alexa Fluor 647-conjugated donkey anti-rabbit (Invitrogen A31573), Alexa Fluor 488-conjugated goat anti-chicken (Invitrogen A11039) and Alexa Fluor 555-conjugated donkey anti-rabbit (Invitrogen A31572) were used for visualizing signals at dilutions of 1:700 (anti-Egr1), 1:2,000 (anti-GFP) and 1:350 (Alexa Fluor antibodies), respectively. Frozen brain sections were incubated in primary and secondary antibodies overnight at 4 °C and for 2 h at room temperature, respectively. For OR expression in the nasal cavity, OE sections were analysed by _in situ_ hybridization[17]. Signals were detected with the Tyramide signal amplification kit (Invitrogen Molecular probes TSA kit 40). All slides were counterstained with 4,6-diamidino-2-phenylindole (DAPI) or Nissl.

**Cell counting.** Images of brain sections stained with anti-Egr1 and DAPI were collected using a fluorescence microscope, Model IX70 (Olympus), coupled to a cooled CCD camera, C4742-95-12ERG (Hamamatsu Photonics). Brain structures were identified microscopically and in digital photos according to the Paxinos and Franklin's mouse brain atlas[33]. To count the number of Egr1$^+$ neurons, Egr1-positive signals were distinguished from background by binarizing the images using a software, Adobe Photoshop.

**Three-dimensional computer graphics for Egr1 expression.** Pictures of fluorescence-labelled Egr1 expression in the 20 μm sections were taken every 100 μm, and volume rendering software (VCAT5), developed by the Image Processing Research Team at RIKEN, was used to incorporate the pictures and to process the images (http://logistics.riken.jp/vcat/vcat/en/). Detailed methodology and procedures for the imaging analysis by VCAT5 have been previously described[34,35]. To adjust the XYZ resolution, pictures with the value of pixel 0 were inserted into the blank space.

**Behavioural analyses of freely moving KI mice.** The 7- to 12-week-old male WT C57BL/6J and ChRWR-Venus KI mice were used for each experiment. At 2 days before the experiment, mice were anaesthetized with ketamine, followed by domitor or isoflurane. The skull above the dorsal OB was carefully thinned not to damage the OB surface with a handheld drill to achieve optical clarity. A custom-built head post was installed to gain access to the top of the skull for photo illumination. For the behavioural experiment, mice were put into a 20 cm × 17 cm transparent box with three cameras (SONY HDR-AS30V) on the top, side and front of the cage. Behaviour was observed for 3 min without photoillumination, and then a 473 nm wavelength of optical stimulation was delivered by an Omicron LuxX laser. A 500 μm plastic optical fibre (LUCIR COME2-DF1-1000) was set to sufficiently illuminate 1.3 mW mm$^{-2}$ light on the dorsal OB. A 2-Hz 250 ms pulse width of 1.3 mW mm$^{-2}$ light stimulation was given for 1 s every 20 s, or for 5 s every minute. As controls, mice whose heads were shielded with aluminum foil to block the light were analysed. Mouse behaviour was analysed with the software, Kinovea, running at a one-fifth reduced speed. AviUtl software was used for video editing. For aversion experiments, KI mice were placed in a 20 cm × 17 cm transparent two-chamber box. Photostimulation of 2 Hz for 1 or 5 s was given every 20 s in one of the two chambers. The time spent in the photostimulation room was measured within 30 min. For immobility experiments, two researchers independently analysed the video in blind and confirmed that the data are over 95% confident.

**Odour-induced behavioural assay of KO mice.** The 7- to 10-week-old KO and WT mice were placed in a 20 cm × 17 cm transparent box and their behaviours were observed for 10 min without odour stimulation. Mice were then given the odour stimulation with 100 μl of 1% or 10% TMT on 2 × 2 cm filter paper. The box was covered with a lid to fill the box with the odourants. For aversive behaviour experiments, mice were placed in 20 cm × 17 cm transparent two-chamber box. A piece of filter paper immersed with 10% TMT was placed in one particular room out of two in a cage. The time spent in the odour-exposure room was measured for the WT and KO mice within a period of 5 min. Data were analysed by GraphPad Prism 4.

**ACTH measurement.** Male WT C57BL/6J and ChRWR-Venus KI mice, 7 to 12 weeks old, were used for each experiment. After an optic fibre was connected to the head post, mice were left to acclimate for 2 h in clean cages. Then, 5 s photo-illumination of a 2 Hz 250 ms pulses with a 10 s interval was repeatedly given for 10 min. For TMT exposure, WT mice were exposed to 10% TMT for 10 min. Mice were immediately decapitated. Plasma ACTH of each sample was analysed by the ACTH EIA kit (MD Bioproduct, cat. no. MO46006) and measured by the Bio-Rad Benchmark Microplate Reader (E15-1233).

**Data availability.** The data that support the findings in this study are available from the corresponding author on request.

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

## Acknowledgements

This research was supported by Specially Promoted Research Grants (Hi.S.) and Grants-in-Aid (Ha.S., H.N., T.K. and K.M.) from the Ministry of Education, Culture, Sports, Science and Technology, Japan, and grants from Urakami Foundation (H.N.) and Hayashi Memorial Foundation for Female Natural Scientist (Ha.S.). This work was also supported by the CREST Program (K.M.) and Japan-Israel Cooperative Scientific Research (Ha.S., K.M. and Hi.S.) sponsored by Japan Science and Technology Agency. We thank T. Ishizuka and H. Yawo (Tohoku U.) for providing us with ChRWR plasmid, A. Miyawaki (RIKEN Inst.) for Venus cDNA, K. Tamura (U. Tokyo) for his advice on laser illumination and S. Itohara (RIKEN Inst.) and his lab members for their advice on the behavioural experiment.

## Author contributions

Ha.S., H.N., T.K., K.M. and Hi.S. conceived and designed the study. Ha.S., H.N., S.S., M.M. and T.Y. performed the experiments. K.M. and T.K. supervised the electrophysiological and behavioural experiments, respectively. Ha.S., H.N. and Hi.S. wrote the paper.

## Additional information

**Competing interests:** The authors declare no competing financial interests.

