## [Peer Review File · Nature Communications]

Reviewers' comments:

Reviewer #1 (Remarks to the Author):

Saito et al investigate the contribution of a single glomerulus in eliciting innate olfactory behavior, namely, innate freezing and avoidance (evoked by TMT) in mice. Specifically they identify an OR with a (moderately) high responsiveness to TMT using an elegant combination of intrinsic signal imaging, RNA sequencing and in vitro imaging screening. They then create mice with ChRWR knocked into the relevant locus for photoactivation of this glomerulus as well as a KO model. They then use these mouse models to investigate physiological responses in principal neurons of the olfactory bulb, characterize odor and light-evoked activity across several brain areas and behavioral responses. This technical tour-de-force manuscript clearly has the potential to have major impact in (and beyond) the field.

To what extent a single glomerulus, as opposed to a pattern of glomerular activation, underlies olfactory behavior is a highly interesting and relevant topic. While Smear et al (2013) have already shown that activation of a single glomerulus can contribute to learned olfactory behavior, the advance in this manuscript that innate behavior, too, can be elicited without learning by a single glomerulus. Overall the current work demonstrated that this is the case: Stimulating a single glomerulus can partially replicate the TMT-evoked phenotype. On the way to this claim the authors make several other analyses that are potentially not essential to this claim - however, some of them have major issues, outlined below. While many of these issues would benefit from additional experiments, it is possible that the authors can address them solely by more detailed analysis and presentation with alterations to the text including refinement of the resulting claims.

(1) The location of KI glomerulus quite different from the general TMT-responsive region; is the wiring disturbed as a result? To claim that a single glomerulus underlies behavior, it is important to establish that that glomerulus is as close to a natural one as possible. This needs to be quantified by comparing positions in the KI and in WT and discussed in detail.

(2) The OB activation pattern from intrinsic signals indicates the relevant area of activation is located dorso-laterally. This likely means that imaging from the dorsal part only cannot cover the whole extent, and as a result probably the ORs screened a subset of TMT-responsive glomeruli. This may explain the fact that the OR they identified is still not as exquisitely tuned to TMT (10% TMT is still a high concentration), even though it was the highest responsive in the OR set they tested. This needs to be discussed

(3) In describing the cortical activation pattern using Egr1 immunohistochemistry, it is unclear why the authors did not concentrate more on the expression patterns in the amygdala which they detail only in the supplement. Apart from the obvious connection to innate fear responses, according to Sosulski et al's work, the projection patterns of M/TCS show specificity most prominently in the amygdala, while the patterns to other cortical regions seem random (so animal-to-animal comparison may be not be very informative). Here, the authors' emphasis seems to be the opposite and it's unclear why. More importantly, however, the Egr1 expression patterns evoked by photoactivation in KI animals look different from those evoked by TMT in WT on first glance at least. It's difficult to judge this, however, because the boundaries of different compartments (white dotted lines in the panels e.g., Supp fig 4-2) are not easily comparable across animals. Authors need to describe in more detail how these borders were determined e.g., what reference structures were used. They also generally need to quantify this part better or significantly reduce its contribution to the manuscript.

As is, it is difficult to understand how similar or dissimilar the TMT or light-evoked responses are - the paper is lacking quantification in this part (with the exception of Supp Fig 4d). Importantly it also lacks a control for "dissimilar pattern" by investigating responses to different odors as well as "Very similar" responses by explicitly comparing Egr1 expression in different animals in responses to TMT (or light). As is, it is very difficult to draw any conclusions from this part.

On a related issue: I find it difficult to find evidence for the following statement in the named

Figure part: "although the TMT-induced activation of the cap compartments of the lateral OT looked weaker in the KO compared with the WT (Supplementary Fig. 4b)." (p11 bottom)
This sounds like very important data and needs to be quantified properly as part of an extended quantification of Fig 4

(4) The difference in putative TC/MC responses to photoactivation of glomerulus may be purely that MCs that belong to the OR-TMT glomerulus are more difficult to record from because of locations. The photo-induced MC responses are too slow to be accounted for even by the indirect transmission described in Gire et al (2012), where, at moderate light intensities, MCs elicit first spikes in less than 100 ms (albeit in vitro). Authors need to detail what fractions of presumptive M/TCs they recorded were categorized as belonging to the OR-TMT glomerulus, and with what measure. In addition, in grouping units into MCL vs EPL units, the method they used to determine the recording locations need more detailing. Presumably, a stimulating electrode was lowered to the LOT a few millimeters below the skull, below the AON/piriform. Such a targeting method is very difficult even with the use of a stereotaxic frame and it is unclear whether the stimulation was exclusively of M/TC axons, or of pyramidal neurons. Both would produce field potentials, but in unknown locations i.e., unknown sink/source locations, and therefore are unlikely to be very reliable as references.

Having said that - the authors (esp Mori) are the experts in the world for this recording / stimulation technique.

Nevertheless, the authors need to elaborate on how they identify a putative MC/TC to belong to the OR-TMT glomerulus. More quantification of the physiology is needed. In particular, the authors need to provide histograms / raw data for all the EPL units, they describe: They comment on the EPL neurons that TMT evoked responses in all photo-activated units. They need to give summary data / histograms for the TMT response for light-activated vs non light-activated EPL units. They need to do the same for all MCL units - i.e. what fraction of light-activated MCL units did show TMT excitation or not. This would boost confidence in the technique and confirm the fraction of light-evoked units that are directly belonging to the OR-TMT glomerulus.

Reiterating what I wrote above - this dataset might not be critical for the paper's main conclusion (that OR-TMT stimulation drives freezing). It is critical, however, for the MC/TC distinction in Egr1 activation and other side aspects of the paper.

(5) The time course of the behavior is difficult to assess. It looks like KI mice are immobile for extended periods of time but that immobility is not necessarily being induced by the individual light stimuli. The shielded control is excellent and convincing that it has to do with the light stimulation. Nevertheless, the responses observed don't look phasic to me. Can the authors repeat these experiments with much longer inter-stimulus-intervals to give the animals enough time to recover their normal mobility? This would convince the reader that immobility is a direct effect of OR-TMT stimulation.

Additionally, a stimulus triggered behavior map would be instructive - i.e. align all trials in all KI mice to the onset of light stimulation and plot e.g. a % trials with immobility, walking, exploring etc.

Minor remarks

(1) Wrong figure referencing (Result section; Photo-induced responses in M/TCs are described in Fig 3, not Fig 2)

(2) abstract l 6 "and" instead of "or"

(3) label J cluster in Fig 1

Difficult to relate Fig 1c to 1b - same order? Mark the ORs used in the different parts?

(4) For most ORs the ORNs project to 2 glomeruli in each bulb. Is OR-TMT an exception? If not - where is the second glomerulus? The authors might want to comment on this.

(5) page 9: they comment on using "250-msec pulses every minute for 20 min, the minimum stimulus needed to induce photo-activation of Egr1 expression"
Is this based on the authors experiments? Can they maybe as a supplement show activation for weaker stimulation to back up this statement?

(6) Supp Fig 4a: What happens in other parts of the OB? Maybe showing more sections would help

(7) The authors should elaborate a bit on the histological definition of the subregions in Fig 4 especially in the OT

(8) on pg 12 - normal respiration in even head fixed mice at rest is 3-4Hz , so 2Hz does not replicate - unnecessary to make claim

(9) pg 13 - make more clear the avoidance experiment is a different experiment to the first one talked about with the freezing responses and define the avoidance behaviour in the text

Reviewer #2 (Remarks to the Author):

In this manuscript, Saito et al investigate the sensory basis of innate fear behavior triggered by the fox odor, TMT. They use an impressively diverse array of techniques to give a comprehensive study of this system. After locating TMT-responsive glomeruli, they determine what odorant receptor genes are expressed in the inputs to these glomeruli. They then determine which of these odorant receptors are most sensitive to TMT. The most sensitive receptor, termed OR^{TMT}, is targeted for knockout, and they made a knockin of ChRWR, with which they can selectively stimulate the OR^{TMT} glomerulus. Using immediate early gene analysis, they survey what olfactory cortical regions are activated by TMT and photostimulation of the OR^{TMT} glomerulus. Lastly, they show that stimulation of the OR^{TMT} glomerulus suffices to evoke freezing, and that deletion of the OR^{TMT} gene reduces fear behavior. All in all, this manuscript is a technical tour de force demonstrating important novel findings. However, I do have a few issues. If these can be resolved, I will support publication in Nature Communications.

1. Figures 1b & 1c don't seem to be consistent with each other. For example, 1b (bottom) shows that Olfr 1023 gives the largest cAMP increase of the screened receptors, yet in 1c it is the weakest-responding D-zone receptor. Further, Olfr 390 gives the second-largest cAMP increase in 1b (bottom), but is absent from Fig 1c. How do you explain these discrepancies?

2. The electrophysiology data in Fig 3 and Supp Fig 3 are too anecdotal. They suffice to show that light can evoke spikes in the OB of KI mice, but the authors make stronger claims -- that longer pulses are required for MCL units to respond, and that light responsive neurons respond to TMT. These should be substantiated in a larger sample of neurons. Fig 3b shows spike rate as a function of light intensity, but only for n=3 neurons. Further, the dependence on stimulus duration is shown for only one EPL and one MCL unit. This is not enough to generalize to the whole population of OR^{TMT}-recipient MT cells. Likewise, only one example neuron is shown in Fig 3d. This is not enough to convincingly demonstrate that light responsive neurons are all TMT responsive.

3. Also on the topic of electrophysiology, in the main text, the authors state "the timing of photo-

illumination induced spikes was independent of inhalation phases". This is simply because the light stimuli were not sniff-triggered and given at specific inhalation phases. This statement is thus misleading.

4. In the section on IEG analysis, the authors say "KI mice were exposed to 2 Hz light in 250-msec pulses every minute for 20 min, the minimum stimulus needed to induce photo-activation of Egr1 expression". In what way is this the minimum? A few paragraphs later, the authors say "we reduced the pulse time from 250 ms to 20 ms," and they still saw Egr1 expression. This contradiction needs to be clarified.

5. In supplementary figure 4-1a, which shows Egr1 expression in the vicinity of the OR⁺TMT glomerulus, does not show the MCL. It's important to know whether and to what extent mitral cells were also activated under these conditions.

6. In the section on IEG analysis, the authors say "In the KO, fear-responsive OC regions and the BNST were activated by TMT (Fig. 4a-c and Supplementary Fig. 4b, e) as in the WT". However, the KO activation is not shown for OC in 4a-b. These data should also be shown.

7. In the section on behavior, the authors say "Intervals of illumination at 2Hz were set to the same frequency as that of the respiration cycle". Does this mean that the light pulses were triggered from the breathing cycle? Or does it mean that 2 Hz is intended as an approximation of the average sniff rate? If the latter, 2 Hz is slower than the average sniff rate of a head-fixed mouse (see eg Verhagen 2007, Shusterman 2011). A freely-moving mouse will sniff even faster. Thus, this statement should be clarified or removed.

8. Also in the behavior section: "It should be noted that aversive responses to TMT were not affected in this KO (Fig. 5f)." Yet figure 5f shows a significant reduction in KO mice aversion. Also, in 5d aversion is quantified in terms of time in the light-on chamber, whereas in 5f it's "distance to TMT". Why the different quantifications?

Reviewer #3 (Remarks to the Author):

The mammalian olfactory system detects countless chemicals perceived as odors. It also detects predator odors that elicit innate fear responses. This suggests that there are genetically defined neural circuits within the olfactory system and olfactory receptors that are involved in innate fear responses to predator odors. In this study, the authors found that Olfr1019, an OR (odorant receptor) that is highly sensitive to a predator odor, TMT, is involved in TMT-induced fear behavior in mice. They found that activation of the olfactory bulb axons of Olfr1019-expressing neurons with optogenetics induced freezing behavior (immobilization), a characteristic fear behavior. Moreover, deletion of the Olfr1019 gene reduced TMT-induced freezing behavior. This is the first paper to show that activation of single OR can induce a specific behavior. This is an important finding that will be of considerable interest to the field. However, I suggest that the paper be revised before publication.

General points:

1. I strongly recommend not to use ORTMT but to use its original name, Olfr1019 in the manuscript. The data presented here suggest that there are other ORs in addition to Olfr1019 that are involved in the fear response to TMT. The name ORTMT may give the wrong impression to readers that Olfr1019 is the only OR involved in TMT-induced fear responses.

2. The authors used 10% TMT for behavioral assays with Olfr1019 KO mice. Since a concentration

of 10% TMT (770mM) is still high, it is likely that many ORs that are less sensitive than Olfr1019 respond to TMT at that concentration. The authors can use a lower concentration of TMT. With that, freezing behavior of Olfr1019 KO mice to TMT might be completely abolished. If so, the lower concentration would also give more informative data about areas of the olfactory cortex activated by Olfr1019, which could be important.

3. The main conclusion of this paper is that Olfr1019 is involved in TMT-induced fear responses. However, the authors include many experiments that analyze the physiological difference between tufted cells and mitral cells downstream of Olfr1019 glomeruli. Since the results in this paper suggest that activation of mitral cells is required for fear responses, I am afraid that this part does not contribute to the paper's conclusions. In fact, it seems rather irrelevant and a distraction. Therefore, the authors should revise the text and make this part shorter or consider eliminating it. It is also unclear from these experiments whether the photostimulation used in behavioral studies actually activated mitral cells-or only tufted cells. This is an important point that the authors need to clarify.

4. The authors analyzed brain areas activated by TMT or photostimulation of Olfr1019 glomeruli by using Egr1 immunostaining. However, they did not draw any conclusions from these data. I am not sure the paper requires these data, particularly without conclusions. At the least, the authors need to improve their data presentation as follows.

i) The authors show only representative drawings of the relative locations of Egr1+ cells in a limited number of sections from olfactory cortex. They need to give quantitative data showing the number of Egr1+ cells in each area in multiple animals. Also, they should include data for the olfactory amygdala in main figures, because it is implicated in innate aversion and freezing behavior to TMT.

ii) The authors need to compare these results with appropriate controls. They need to analyze the number of Egr1+ cells in mice exposed to water, and in M72-ChR2 mice (used in this paper) stimulated with light.

iii) It would be extremely helpful if the authors included analysis of KO mice in all analyzed brain areas. This could be an important addition to the paper.

5. The use of the term "glomerulus module" is rather confusing because it is unclear what the authors mean by it. If the authors mean a glomerulus, they should simply use the term "glomerulus". If they mean something else, they should define what it is.

6. The authors should mention that the innate fear response to TMT includes not only behaviors (avoidance, freezing), but also an increase in the blood stress hormone, ACTH (and corticosterone). One important finding here is that stimulation of Olfr1019 glomeruli induced freezing, but not aversion. Interestingly, a previous paper showed that one part of the olfactory amygdala (AmPir) is involved in the stress hormone response to TMT (Kondoh et al, 2016), but not freezing behavior, indicating an additional dissociation between different parts of the fear response at the level of the cortex.

7. For clarity, the authors should avoid using the term "central brain" to describe the olfactory cortex. If they mean the olfactory cortex and the BNST, it would be better to say that. If the authors use the term central brain, they need to define it.

Specific points:

1. To identify ORs that respond to TMT, the authors injected DiI into activated glomeruli identified by optical imaging. However, although they found 2-4 glomeruli activated by 10% TMT, they found 62 ORs labeled by DiI and one gets the impression (although it is unclear) that many of those ORs do not recognize TMT. This suggests the possibility that the DiI injections are not accurate and may result in DiI entry into other glomeruli near the one targeted. The authors should include data showing the quality of DiI injection and how many glomeruli contained DiI after injection. Fig. S1b

is not enough. It is also important that the authors provide more information on the other ORs that were identified. Why were only thirteen of them analyzed for TMT responses in HEK cells? One gets the impression that criteria were used, but what those criteria were is unclear. It would also help to provide more clearcut information about the number of animals and glomeruli that gave individual ORs in the supplementary table.

2. In Fig. 1d, the annotation of the table should be revised.

3. The authors should use the term "olfactory amygdala" instead of "amygdala" to describe amygdala areas located in the olfactory cortex.

4. In Fig.5 and Fig. S5, the authors used the T-test, and compared multiple groups for statistical significance. However, it is not clear whether they adopted corrections for multiple comparisons, such as Bonferroni correction. Also, it is not clear whether all groups are compared, or only some groups are compared.

5. In Fig. S5d, the basal ACTH concentration is extremely high compared with published results (e.g. Kobayakawa et al. Nature 2007, Papes et al. Cell 2010, Kondoh et al. Nature 2016). Did the authors calculate the ACTH concentration correctly?

6. p.10. In contrast to the authors' statement, I don't think there is any evidence that the PPC is involved in inducing stress responses. This should be corrected. In the following sentence, the AmPir should be mentioned for its role in stress hormone responses to predator odors, including TMT (Kondoh et al, 2016)

7. There are several mistakes in figure citations on page 8. All Fig.2b should be Fig.3b.

8. Several citations for previous publications should be revised.

i) On page 3, the authors wrote, "More recently, a group of OR genes named TAARs have been identified for trace amine-associated receptors whose glomeruli have been mapped to a new subdomain, DIII 16-18." They should include "Pacifco et al. 2012 Cell", because Tom Bozza's group mentioned DIII domain first in this paper. Second, TAARs are not ORs (odorant receptors) as implied in this statement.

ii) On page 3, the authors wrote, "Odor information encoded in the glomerular map in the OB is further processed by local circuits and conveyed by mitral and tufted (M/T) cells to various areas of the olfactory cortex (OC)". They should include one more paper published with Miyamichi et al. and Sosulski et al. in the same journal: Ghosh et al. 2011 Nature.

iii) On page 6, the authors wrote, "This OR is in a list of TMT-responsive ORs recently identified by the in vivo screening³⁰." However, TMT was not tested in the paper (von der Weid et al. Nature Neurosci. 2015). In a different paper (Jiang et al. Nature Neurosci. 2015), the authors did test TMT, but did not find Olfr1019 as a TMT-responsive OR.

iv) On page7, the authors wrote, "This situation is quite different from that reported for the TAAR2 KO mice in which OSNs expressing the KO allele always chose another TAAR gene from the same gene cluster¹⁸." The cited paper (Dewan et al. Nature 2013) did not analyze expression of TAARs in TAAR KO OSNs. The authors should cite different papers, Pacifco et al. 2012 Cell Reports, Johnson et al. 2012 PNAS, and Yoon et al. 2015 PNAS. In addition, no study used "TAAR2" KO mice. Finally, all TAAR genes are located in a single cluster, so the statement cannot correct.

We are pleased to find that overall comments from all three reviewers are quite positive and would like to thank them for their constructive criticisms and suggestions. We greatly appreciate their time and effort in reading our manuscript and raising various points. We acknowledge that these comments were quite helpful in improving our paper. Although the time for revision was limited, we made every our effort to respond to the reviewers' comments and to address their questions.

Point-by-point responses to each reviewer are as follows (reviewers' comments are listed in bold-faced):

Reviewer #1:

Saito et al investigate the contribution of a single glomerulus in eliciting innate olfactory behavior, namely, innate freezing and avoidance (evoked by TMT) in mice. Specifically they identify an OR with a (moderately) high responsiveness to TMT using an elegant combination of intrinsic signal imaging, RNA sequencing and in vitro imaging screening. They then create mice with ChRWR knocked into the relevant locus for photoactivation of this glomerulus as well as a KO model. They then use these mouse models to investigate physiological responses in principal neurons of the olfactory bulb, characterize odor and light-evoked activity across several brain areas and behavioral responses. This technical tour-de-force manuscript clearly has the potential to have major impact in (and beyond) the field.

To what extent a single glomerulus, as opposed to a pattern of glomerular activation, underlies olfactory behavior is a highly interesting and relevant topic. While Smear et al (2013) have already shown that activation of a single glomerulus can contribute to learned olfactory behavior, the advance in this manuscript that innate behavior, too, can be elicited without learning by a single glomerulus. Overall the current work demonstrated that this is the case: Stimulating a single glomerulus can partially replicate the TMT-evoked phenotype. On the way to this claim the authors make several other analyses that are potentially not essential to this claim - however, some of them have major issues, outlined below. While many of these issues would benefit from additional experiments, it is possible that the authors can address them solely by more detailed analysis and presentation with alterations to the text including refinement of the resulting claims.

(1) The location of KI glomerulus quite different from the general TMT-responsive region; is the wiring disturbed as a result? To claim that a single glomerulus underlies behavior, it is important to establish that that glomerulus is as close to a natural one as possible. This needs to be quantified by comparing positions in the KI and in WT and discussed in detail.

We agree that this is an important point. Responding to the reviewer's suggestion, we quantified the two sets of data for glomerular locations: one set was obtained from fluorescent labeling of the KI, and the other was from optical imaging of the WT. Locations of Venus-positive glomeruli in the KI were found to be well-correlated with those of TMT-responsive glomeruli in the WT. These data are now shown in new Supplementary Fig. 2c and mentioned in the text (page 5, line 18 - page 5, line 24).

(2) The OB activation pattern from intrinsic signals indicates the relevant area of activation is located dorso-laterally. This likely means that imaging from the dorsal part only cannot cover the whole extent, and as a result probably the ORs screened a subset of TMT-responsive glomeruli. This may explain the fact that the OR they identified is still not as exquisitely tuned to TMT (10% TMT is still a high concentration), even though it was the highest responsive in the OR set they tested. This needs to be discussed

TMT activates multiple glomeruli in both the dorsal and ventral OB regions. We have previously reported that the glomeruli responsible for inducing innate fear are confined to the D_{II} subdomain of the mouse OB. Furthermore, in the lateral olfactory map, TMT-responsive glomeruli are clustered in the dorsolateral region (Kobayakawa *et al.*, *Nature*, **450**, 503-508, 2007). We, therefore, analyzed the dorsolateral OB for TMT-responsive glomeruli by optical imaging and identified Olfr1019 as one of the promising candidates. However, TMT-responsive glomeruli are also present in the dorsomedial region of the medial map (Kobayakawa *et al.*, *Nature*, **450**, 503-508, 2007). As the reviewer points out, there is a possibility that we might have missed better ORs that were not covered in our imaging experiment. We, therefore, performed high-throughput screening of 266 OR clones and identified additional TMT-responsive ORs (Supplementary Fig. 1d). After the luciferase assay of all candidate clones (Fig. 1b and c), Olfr1019 was still found to be the best OR and thus used for further studies. This is now mentioned in the text (page 4, line 3 - page 5, line 7).

(3) In describing the cortical activation pattern using Egr1 immunohistochemistry, it is unclear why the authors did not concentrate more on the expression patterns in the amygdala which they detail only in the supplement. Apart from the obvious connection to innate fear responses, according to Sosulski et al's work, the projection patterns of M/TCS show specificity most prominently in the amygdala, while the patterns to other cortical regions seem random (so animal-to-animal comparison may be not be very informative). Here, the authors' emphasis seems to be the opposite and it's unclear why. More importantly, however, the Egr1 expression patterns evoked by photoactivation in KI animals look different from those evoked by TMT in WT on first glance at least. It's difficult to judge this, however, because the boundaries of different compartments (white dotted lines in the panels e.g., Supp fig 4-2) are not easily comparable across animals. Authors need to describe in more detail how these borders were determined e.g., what reference structures were used. They also generally need to quantify this part better or significantly reduce its contribution to the manuscript.

As is, it is difficult to understand how similar or dissimilar the TMT or light-evoked responses are - the paper is lacking quantification in this part (with the exception of Supp Fig 4d). Importantly it also lacks a control for "dissimilar pattern" by investigating responses to different odors as well as "Very similar" responses by explicitly comparing Egr1 expression in different animals in responses to TMT (or light). As is, it is very difficult to draw any conclusions from this part.

On a related issue: I find it difficult to find evidence for the following statement in the named Figure part: "although the TMT-induced activation of the cap compartments of the lateral OT looked weaker in the KO compared with the WT (Supplementary Fig. 4b)." (p11 bottom)

This sounds like very important data and needs to be quantified properly as part of an

extended quantification of Fig 4

According to the reviewer's suggestions, we quantified activation levels by counting the Egr1-positive neurons. In this experiment, we analyzed the photo-illuminated Olfr1019-ChRWR-KI, photo-illuminated M72-ChR2-YFP, TMT-exposed WT, TMT-exposed Olfr1019 KO, and vanillin-exposed WT mice. Positions of brain slices were determined based on the distance from the bregma referring Paxinos and Franklin's mouse brain atlas. The samples were counterstained with DAPI (Supplementary Fig. 4c-h) or Nissl. Egr1-positive neurons were counted in the AON, OT, and CoA (Fig. 4), concentrating on the amygdala as advised by the reviewer. Quantification procedures are described in the Methods section (page 16, line 9-15). Additional data for the OB, AON, APC, OT, CoA, MeA, and BNST are also shown in Fig. 4a-c and Supplementary Fig. 4b-h. As negative controls, the photo-illuminated and mineral oil-exposed WT mice were analyzed, whose data are included in Supplementary Fig. 4c-h. These analyses are mentioned in the text (page 8, line 3 - page 9, line 6).

(4) The difference in putative TC/MC responses to photoactivation of glomerulus may be purely that MCs that belong to the ORTMT glomerulus are more difficult to record from because of locations. The photo-induced MC responses are too slow to be accounted for even by the indirect transmission described in Gire et al (2012), where, at moderate light intensities, MCs elicit first spikes in less than 100 ms (albeit in vitro). Authors need to detail what fractions of presumptive M/TCs they recorded were categorized as belonging to the ORTMT glomerulus, and with what measure. In addition, in grouping units unto MCL vs EPL units, the method they used to determine the recording locations need more detailing. Presumably, a stimulating electrode was lowered to the LOT a few millimeters below the skull, below the AON/piriform. Such a targeting method is very difficult even with the use of a stereotaxic frame and it is unclear whether the stimulation was exclusively of M/TC axons, or of pyramidal neurons. Both would produce field potentials, but in unknown locations i.e., unknown sink/source locations, and therefore are unlikely to be very reliable as references. Having said that - the authors (esp Mori) are _the_ experts in the world for this recording / stimulation technique.

Nevertheless, the authors need to elaborate on how they identify a putative MC/TC to belong to the OR-TMT glomerulus. More quantification of the physiology is needed. In particular, the authors need to provide histograms / raw data for all the EPL units, they describe: They comment on the EPL neurons that TMT evoked responses in all photo-activated units. They need to give summary data / histograms for the TMT response for light-activated vs non light-activated EPL units. They need to do the same for all MCL units - i.e. what fraction of light-activated MCL units did show TMT excitation or not. This would boost confidence in the technique and confirm the fraction of light-evoked units that are directly belonging to the ORTMT glomerulus.

Reiterating what I wrote above - this dataset might not be critical for the paper's main conclusion (that OR-TMT stimulation drives freezing). It is critical, however, for the MC/TC distinction in Egr1 activation and other side aspects of the paper.

As for the onset of M/T-cell spike responses, the data of Gire *et al.* (2012) cannot be simply compared with our electro-physiological data: Gire *et al.* used the voltage patch-clamp method for recording the postsynaptic EPSCs of M/T cells within a single glomerulus, while

we employed the extracellular recording method that detect action potentials near the cell bodies of M/T cells. Furthermore, conditions for photo-illumination are also quite different: Gire *et al.* expressed the excess amount of ChR2 in OSNs using the *Tet* promotor and photo-illuminated glomeruli with strong light (1-2 msec pulses, 15-20mW power), while we expressed ChRWR in OSNs using the *Olf1019* promotor and illuminated the Olfr1019 glomeruli with moderate illumination condition (20-400 msec, 1.3 mW power). Since Gire *et al.* described in their paper that “the mean onset time of the light-evoked MC current was faster than that of the electrical stimulation”, their illumination condition seems to be too strong to be physiological. The timing of odor-evoked M/T-cell activity has been reported by Igarashi *et al.* (*J. Neurosci.* **32**, 7970-7985, 2012), showing the difference in the onsets of action potentials between the mitral cells and tufted cells. Comparing our results with those of Igarashi *et al.*, our illumination conditions seem to be more physiological.

We added the LFP results of LOT-stimulated M/T cells in Supplementary Fig. 3b. Our data demonstrate that the configuration and polarity of the field potentials change when the tip of microelectrode is moved to different depths. For setting up the LOT stimulation, concentric electrodes were inserted into the LOT ~2.0 mm anterior to the bregma, 2.3 mm lateral from the midline, and 3.0~5.5 mm from the brain surface. We confirmed LOT stimulation by LFP recording in the GL 500-600 μ l from the surface of the OB. We detected positive waves 100 μ sec after the LOT stimulation in the GCL. We then performed recording in the upper positions in the OB and examined if spike shapes would switch from positive to negative in the EPL, and if spikes would become the smallest in the MCL (Supplementary Fig. 3b) (*J. Physiol.* **168**, 65-88, 1963 and *J. Neurophys.* **101**, 1890-1900, 2009). We kept moving the locations of electrodes until we obtained proper LFP responses in the OB.

In order to identify the photo-activated M/T cells, we performed extracellular recording many times. We examined neighboring cells right underneath the Venus-positive glomeruli and identified one photo-activated MCL or EPL unit after 10-60 trials. Since no cells were photo-activated in the WT mice, we assume that M/T cells evoked by photo-illumination are likely to be connected with the Olfr1019-ChRWR glomeruli.

We also added detailed electro-physiology data in Fig. 3c showing firing frequencies in the EPL and MCL units responding to the different intensities of photo-stimulation. In these experiments, the EPL unit generated initial burst discharges of over 100 Hz. In contrast, the MCL unit yielded burst discharges of 50 Hz (Fig. 3b and c, Supplementary Fig. 3c). These results are consistent with our previous studies demonstrating action potentials of M/T cells induced by odor stimulation (*J. Neurosci.* **32**, 7970-7985, 2012). For more details, please see the attached Physiology data 1-5. We also included the data of non-photo-evoked M/T cells in Physiology data 3 for comparison.

We removed Fig. 3d from the previous manuscript, as the data were preliminary for both photo-activated and TMT-stimulated EPL and MCL units. We included these data in Physiology data 6, as we analyzed only one example of a single EPL unit. Although we wanted to pursue these experiments further, Prof. Kensaku Mori's laboratory was closed due to his retirement: For setting up the electrophysiological experiments in our own lab, it would easily take another year or two. Thus, we minimized the description of M/T-cell, as the main purpose of these electrophysiological experiments was to optimize the illumination condition for *Egr1* experiments and behavioral studies. Responding to the reviewer's point, we described detailed experimental procedure for the identification of photo-activated M/T cells (page 14, line 14-page 15, line 6).

(5) The time course of the behavior is difficult to assess. It looks like KI mice are immobile for extended periods of time but that immobility is not necessarily being induced by the individual light stimuli. The shielded control is excellent and convincing that it has to do with the light stimulation. Nevertheless, the responses observed don't look phasic to me. Can the authors repeat these experiments with much longer inter-stimulus-intervals to give the animals enough time to recover their normal mobility? This would convince the reader that immobility is a direct effect of OR-TMT stimulation.

Additionally, a stimulus triggered behavior map would be instructive - i.e. align all trials in all KI mice to the onset of light stimulation and plot e.g. a % trials with immobility, walking, exploring etc.

According to the reviewer's suggestion, we plotted frequencies of various mouse behaviors induced by photo-illumination. This is now included in new figures (Fig. 5b and Supplementary Fig. 5a, b) and mentioned in the text (page 9, line 21-page 10, line 6). Photo-induced immobility responses often lasted longer than 60 sec. Responding to the reviewer's comment, we added the time course of various behaviors for extended time periods both before and after stimulations (Fig. 5a).

Minor remarks

(1) Wrong figure referencing (Result section; Photo-induced responses in M/TCs are described in Fig 3, not Fig 2)

Yes, it is wrong as pointed out. This has been corrected.

(2) abstract l 6 "and" instead of "or"

This has been corrected in the new abstract.

(3) label J cluster in Fig 1

Difficult to relate Fig 1c to 1b - same order? Mark the ORs used in the different parts?

We made necessary changes in Supplementary Fig. 1d for improvement.

(4) For most ORs the ORNs project to 2 glomeruli in each bulb. Is OR-TMT an exception? If not - where is the second glomerulus? The authors might want to comment on this.

As usually found for other OR species, Olfr1019 gives rise to a pair of glomeruli, one is in the medial and the other is in the dorsal part of the OB. This is clearly shown in Fig. 2c and Supplementary Figs. 2d and 4b.

(5) page 9: they comment on using "250-msec pulses every minute for 20 min, the minimum stimulus needed to induce photo-activation of Egr1 expression" Is this based on the authors experiments? Can they maybe as a supplement show

activation for weaker stimulation to back up this statement?

The illumination condition of 250 msec pulses every minute for 20 min was not correct. The condition actually used was 250 msec pulses with 250 msec intervals (2Hz) for 30 min. We chose to use this condition because clear Egr1 signals were obtained consistently in the OB, OC, MeA, and BNST. This condition was used for quantifying Egr1 signals in these brain regions. In response to the reviewer's comment, we added a new set of Egr1 data with the photo-illumination condition of 20 msec pulses 2Hz for 30 min, and 250 msec pulses 2 Hz for 15 min (Supplementary Fig. c-h). With the 20 msec stimulation, Egr1 signals were too faint to be counted. With the illumination of 250 msec pulses 2 Hz for 15 min, the results were similar to those with the stimulation for 30 min, but the signals were much weaker in the MeA and BNST.

(6) Supp Fig 4a: What happens in other parts of the OB? Maybe showing more sections would help

As advised by the reviewer, more OB sections were analyzed in the broader field. New data are included in Supplementary Fig.4b, and also mentioned in the text (page 8, line 6-10).

(7) The authors should elaborate a bit on the histological definition of the subregions in Fig 4 especially in the OT

As for the histological definition, we made necessary changes in the text (page 8, line 16-19), and in Supplementary Fig. 4d with DAPI staining.

(8) on pg 12 - normal respiration in even head fixed mice at rest is 3-4Hz , so 2Hz does not replicate - unnecessary to make claim

We agree with the reviewer. Descriptions and figures for respiration/inhalation were removed. For the illumination condition, we used the 250 msec pulse for behavioral experiments, because it induced the plateau in the spike-discharge rate of M/T cells. Since we detected prolonged action potentials in the MCL unit after giving a pulse of 250 msec, we put enough intervals (250 msec) to generate on-off firings in the MCL unit. We also examined 20 msec 5 Hz in extracellular recording and found the slower onset and lower frequency of firing in the MCL unit (Physiology data 2).

(9) pg 13 - make more clear the avoidance experiment is a different experiment to the first one talked about with the freezing responses and define the avoidance behaviour in the text

Responding to the reviewer's comment, we performed a new set of avoidance experiments and added the results in Fig. 5d and f. These are mentioned in the Methods (page 17, line 16-20, and page 18, line 1-6) and the legend to Fig. 5d and f.

Reviewer #2:

In this manuscript, Saito et al investigate the sensory basis of innate fear behavior

triggered by the fox odor, TMT. They use an impressively diverse array of techniques to give a comprehensive study of this system. After locating TMT-responsive glomeruli, they determine what odorant receptor genes are expressed in the inputs to these glomeruli. They then determine which of these odorant receptors are most sensitive to TMT. The most sensitive receptor, termed OR^{TMT}, is targeted for knockout, and they made a knockin of ChRWR, with which they can selectively stimulate the OR^{TMT} glomerulus. Using immediate early gene analysis, they survey what olfactory cortical regions are activated by TMT and photostimulation of the OR^{TMT} glomerulus. Lastly, they show that stimulation of the OR^{TMT} glomerulus suffices to evoke freezing, and that deletion of the OR^{TMT} gene reduces fear behavior. All in all, this manuscript is a technical tour de force demonstrating important novel findings.

However, I do have a few issues. If these can be resolved, I will support publication in Nature Communications.

1. Figures 1b & 1c don't seem to be consistent with each other. For example, 1b (bottom) shows that Olfr 1023 gives the largest cAMP increase of the screened receptors, yet in 1c it is the weakest-responding D-zone receptor. Further, Olfr 390 gives the second-largest cAMP increase in 1b (bottom), but is absent from Fig 1c. How do you explain these discrepancies?

High-throughput screening was performed as a backup to cover all areas in the OB. Since this screening was not so precise as DiI screening, we reexamined candidate ORs for their TMT-reactivities by the luciferase assay (Fig. 1b). To clarify the point by the reviewer, we described the detailed experimental procedures for cDNA screening in the legends to Fig. 1b and Supplementary Fig. 1d. Supplementary Table was also modified for better understanding.

2. The electrophysiology data in Fig 3 and Supp Fig 3 are too anecdotal. They suffice to show that light can evoke spikes in the OB of KI mice, but the authors make stronger claims -- that longer pulses are required for MCL units to respond, and that light responsive neurons respond to TMT. These should be substantiated in a larger sample of neurons. Fig 3b shows spike rate as a function of light intensity, but only for n=3 neurons. Further, the dependence on stimulus duration is shown for only one EPL and one MCL unit. This is not enough to generalize to the whole population of OR^{TMT}-recipient MT cells. Likewise, only one example neuron is shown in Fig 3d. This is not enough to convincingly demonstrate that light responsive neurons are all TMT responsive.

As pointed out, we admit that some data were sometimes insufficient. Although the time was limited, we tried our best in improving the data. New electrophysiological data for the EPL and MCL units were added in Fig. 3c, Supplementary Fig. 3c-e, and Physiology data 1-6. We carefully examined the locations of recording glass-tips by measuring the depth and by determining the layer using LFP recording. LOT stimulation was employed for the activation of M/T cells as shown in Supplementary Fig. 3b. This is also mentioned in the text (page 6, line 18-20).

In order to detect the photo-activated M/T cells, we performed extracellular recording many times. We examined neighboring cells right underneath the ChRWR-positive glomeruli and found photo-activated M/T cells after 10-60 trials. In these experiments, we

found that the light intensity over $1\text{mW}/\text{mm}^2$ with a pulse duration of 200 ms induced the plateau in the spike rate in the EPL and MCL units measured by extracellular single-unit recording. We added new Egr1 results of KI mice photo-illuminated with “20 msec” pulses and 250 msec intervals (Supplementary Fig. b-h). In the 20 msec stimulation, Egr1 signals were faint and not convincing. In the behavioral experiments, we used different pulse lengths (20, 50 100, and 250 msec) and the lower intensity ($0.3\text{ mW}/\text{mm}^2$) of light exposure (Supplementary Fig. 5a and e). The shorter pulse length or lower light intensity did not induce notable immobility responses in the KI mice. These results are very well correlated with the results of extracellular recordings. As the main purpose of our electrophysiological experiments was to determine the optimum photo-illumination condition for Egr1 expression and for behavioral responses in the KI mice, we eliminated insufficient data from the manuscript.

In addition to the electrophysiological experiments, we examined Egr1 expression in the OB (Supplementary Fig. 4b). TMT-exposure broadly activated various cells, however, photo-illumination activated only a limited number of cells in the vicinity of Olfr1019 glomeruli. These results indicate that photo-illumination induces the activation of neurons connecting with the Olfr1019 glomerulus.

3. Also on the topic of electrophysiology, in the main text, the authors state "the timing of photo-illumination induced spikes was independent of inhalation phases". This is simply because the light stimuli were not sniff-triggered and given at specific inhalation phases. This statement is thus misleading.

Since these descriptions were misleading as pointed out, we removed them from the text.

4. In the section on IEG analysis, the authors say "KI mice were exposed to 2 Hz light in 250-msec pulses every minute for 20 min, the minimum stimulus needed to induce photo-activation of Egr1 expression". In what way is this the minimum? A few paragraphs later, the authors say "we reduced the pulse time from 250 ms to 20 ms," and they still saw Egr1 expression. This contradiction needs to be clarified.

We chose to use the illumination condition of 250 msec pulses with 250 msec intervals for 30 min. A critical thing in setting the best illumination condition was to obtain the reproducible Egr1 activation in different experiments. Our illumination condition mentioned above produced the Egr1 activation pattern more constantly than the shorter photo-illumination condition. We now included new Egr1 data using the photo-illumination conditions of “20 msec” pulses with 250 msec intervals for 30 min and 250 msec pulses with 250 msec intervals for “15 min” (Supplementary Fig. c-h). With the 20 msec stimulation, Egr1 signals were faint and difficult to detect. In the experiment using the illumination condition of 250 msec pulses with 250 msec intervals for 15 min, the result was similar to that with the stimulation for 30 min. However, the Egr1 signals were much weaker in the MeA and BNST.

5. In supplementary figure 4-1a, which shows Egr1 expression in the vicinity of the OR^TMT glomerulus, does not show the MCL. It's important to know whether and to what extent mitral cells were also activated under these conditions.

To address the reviewer’s concern, we included new photos with broader OB regions in Supplementary Fig. 4b. In the photo-illuminated KI mice, Egr1 signals were found in

periglomerular cells (PGCs) only within the *Olf1019* glomeruli as well as in some mitral cells (MCs) right underneath. In contrast, in the TMT-exposed WT mice, *Egr1*-positive MCs and PGCs were detected in the broader region of the OB. These observations are now mentioned in the text (page 8, line 6-10), and the data are included in Supplementary Fig. 4b.

6. In the section on IEG analysis, the authors say "In the KO, fear-responsive OC regions and the BNST were activated by TMT (Fig. 4a-c and Supplementary Fig. 4b, e) as in the WT". However, the KO activation is not shown for OC in 4a-b. These data should also be shown.

As pointed out, it is important to analyze the TMT-exposed KO mice for *Egr1* expression in the OC. To address the reviewer's point, we analyzed *Egr1* expression in the AON, OT, CoA, and MeA of the KO mice (Fig. 4a-c, and Supplementary Fig. 4b-f). These new data are now mentioned in the text (page 8, line 3-page 9, line 6).

7. In the section on behavior, the authors say "Intervals of illumination at 2Hz were set to the same frequency as that of the respiration cycle". Does this mean that the light pulses were triggered from the breathing cycle? Or does it mean that 2 Hz is intended as an approximation of the average sniff rate? If the latter, 2 Hz is slower than the average sniff rate of a head-fixed mouse (see eg Verhagen 2007, Shusterman 2011). A freely-moving mouse will sniff even faster. Thus, this statement should be clarified or removed.

We totally agree with the reviewer. We tried to find the most effective illumination condition for M/T cell activation. We chose to use the 250 msec pulse for our behavioral experiments, because this condition induced the plateau in the spike-discharge rate of M/T cells (Fig.3c). Since we detected prolonged action potentials in the MCL unit after giving a pulse of 250 msec, we put enough intervals (250 msec) to generate on-off firings in the MCL unit. We also examined 20 msec 5 Hz in extracellular recording, where the slower onset and lower frequency of firing was found in the MCL unit (Physiology data 2).

8. Also in the behavior section: "It should be noted that aversive responses to TMT were not affected in this KO (Fig. 5f)." Yet figure 5f shows a significant reduction in KO mice aversion. Also, in 5d aversion is quantified in terms of time in the light-on chamber, whereas in 5f it's "distance to TMT". Why the different quantifications?

In the previous experiment that measured the distance from odor, it appeared that there was a difference between the WT and KO as noticed by the reviewer. We, therefore, performed a new set of experiments to re-examine the avoidance behavior. We measured the time duration staying in the room where the odor was presented. In this avoidance test, no significant difference was found between the WT and KO (P values $>.9999$). These data are now included in Fig. 5f and mentioned in the text (page 11, line 6-7). Procedures for this avoidance test are described in the Methods (page 17, line 16-20, page 18, line 1-6) and in the legend to Fig. 5f. We also analyzed the previous data by measuring the distance from odor, using Bonferroni correction. P value was found to be 0.3433.

Reviewer #3:

The mammalian olfactory system detects countless chemicals perceived as odors. It also detects predator odors that elicit innate fear responses. This suggests that there are genetically defined neural circuits within the olfactory system and olfactory receptors that are involved in innate fear responses to predator odors. In this study, the authors found that Olfr1019, an OR (odorant receptor) that is highly sensitive to a predator odor, TMT, is involved in TMT-induced fear behavior in mice. They found that activation of the olfactory bulb axons of Olfr1019-expressing neurons with optogenetics induced freezing behavior (immobilization), a characteristic fear behavior. Moreover, deletion of the Olfr1019 gene reduced TMT-induced freezing behavior. This is the first paper to show that activation of single OR can induce a specific behavior. This is an important finding that will be of considerable interest to the field. However, I suggest that the paper be revised before publication.

General points:

1. I strongly recommend not to use ORTMT but to use its original name, Olfr1019 in the manuscript. The data presented here suggest that there are other ORs in addition to Olfr1019 that are involved in the fear response to TMT. The name ORTMT may give the wrong impression to readers that Olfr1019 is the only OR involved in TMT-induced fear responses.

As pointed out, it is true that there are many OR species that respond to TMT. We, therefore, changed the name of OR-TMT to Olfr1019 throughout the manuscript.

2. The authors used 10% TMT for behavioral assays with Olfr1019 KO mice. Since a concentration of 10% TMT (770mM) is still high, it is likely that many ORs that are less sensitive than Olfr1019 respond to TMT at that concentration. The authors can use a lower concentration of TMT. With that, freezing behavior of Olfr1019 KO mice to TMT might be completely abolished. If so, the lower concentration would also give more informative data about areas of the olfactory cortex activated by Olfr1019, which could be important.

We totally agree with the reviewer. This is an interesting possibility. We, therefore, analyzed lower TMT concentrations for the intrinsic signals in the dorsal OB (Fig. 1a). Contrary to our expectation, lowering the TMT concentrations simply decreased signal intensities of 3-4 activated glomeruli rather than narrowing the number down to one particular glomerulus, hopefully for Olfr1019. We also examined fear responses and Egr1 expression in the OC using 1% TMT. By lowering TMT concentrations, the difference in immobility responses became clearer between the KO and WT (Supplementary Fig. 5f). Unfortunately, however, lowering TMT concentrations further did not separate immobility and aversion. As for the Egr1 expression, 1% TMT gave just weaker signals than 10% TMT, but did not make the distribution patterns simpler, or similar to those found in the photo-illuminated KI mice. However, as 1% TMT gave lower background for Egr1 than 10% TMT, the differences in the Egr1 distribution became clearer between the KO and WT (Fig. 4a-c and Supplementary Fig. 4c-h).

3. The main conclusion of this paper is that Olfr1019 is involved in TMT-induced fear responses. However, the authors include many experiments that analyze the

physiological difference between tufted cells and mitral cells downstream of Olfr1019 glomeruli. Since the results in this paper suggest that activation of mitral cells is required for fear responses, I am afraid that this part does not contribute to the paper's conclusions. In fact, it seems rather irrelevant and a distraction. Therefore, the authors should revise the text and make this part shorter or consider eliminating it. It is also unclear from these experiments whether the photostimulation used in behavioral studies actually activated mitral cells-or only tufted cells. This is an important point that the authors need to clarify.

As also mentioned by other reviewers, this reviewer points out that these M/T-cell data do not contribute to the paper's conclusion. We agree to the reviewers' point. We originally performed these experiments to obtain the optimum illumination conditions for Egr1 activation in the OC and for behavioral responses. We, therefore, removed irrelevant or incomplete descriptions of M/T-cell electrophysiology and related figures.

4. The authors analyzed brain areas activated by TMT or photostimulation of Olfr1019 glomeruli by using Egr1 immunostaining. However, they did not draw any conclusions from these data. I am not sure the paper requires these data, particularly without conclusions. At the least, the authors need to improve their data presentation as follows.

- i) The authors show only representative drawings of the relative locations of Egr1+ cells in a limited number of sections from olfactory cortex. They need to give quantitative data showing the number of Egr1+ cells in each area in multiple animals. Also, they should include data for the olfactory amygdala in main figures, because it is implicated in innate aversion and freezing behavior to TMT.**
- ii) The authors need to compare these results with appropriate controls. They need to analyze the number of Egr1+ cells in mice exposed to water, and in M72-ChR2 mice (used in this paper) stimulated with light.**
- iii) It would be extremely helpful if the authors included analysis of KO mice in all analyzed brain areas. This could be an important addition to the paper.**

The reviewer recommends to show the quantitative data for Egr1-positive cells and also to include the data for the olfactory amygdala in the main figure. According to the reviewer's suggestion, we quantified the Egr1-positive neurons in the AON, OT, CoA, and MeA, whose quantification data are now shown in Fig. 4 and Supplementary Fig. 4c-g.

In order to define the brain regions specifically activated in the photo-illuminated KI mice, the reviewer recommended to compare the KI mouse results with those of the Olfr1019-KO and M72-Ch2-YFP mice. These data together with those of vanillin-exposed WT mice are now shown in Fig.4a-c and Supplementary Fig. 4c-h. It was revealed that Egr1 signals are more prominent in the photo-illuminated KI mice than the controls in the following regions:

1. In the AON, Egr1 signals were notably high in the posterior-dorsal region (Fig. 4a and Supplementary Fig. 4c);
2. In the OT, the anterior part of cap compartments of the lateral OT was intensively labeled (Fig. 4b and Supplementary Fig. 4e);
3. In the CoA and MeA, more Egr1 signals were detected in the anterior part (Fig. 4c and Supplementary Fig. 4f, g).

In contrast, the signals were reduced in these areas in the TMT-exposed KO mice (Fig.4a-c and Supplementary Fig. 4c-h).

5. The use of the term "glomerulus module" is rather confusing because it is unclear what the authors mean by it. If the authors mean a glomerulus, they should simply use the term "glomerulus". If they mean something else, they should define what it is.

Since the reviewer finds this term confusing, it was changed to “glomerulus” throughout the manuscript.

6. The authors should mention that the innate fear response to TMT includes not only behaviors (avoidance, freezing), but also an increase in the blood stress hormone, ACTH (and corticosterone). One important finding here is that stimulation of Olfr1019 glomeruli induced freezing, but not aversion. Interestingly, a previous paper showed that one part of the olfactory amygdala (AmPir) is involved in the stress hormone response to TMT (Kondoh et al, 2016), but not freezing behavior, indicating an additional dissociation between different parts of the fear response at the level of the cortex.

The reviewer suggests us to describe the ACTH levels in the KI after photo-illumination. We agree that this is an important point. In our experiments, the ACTH levels were significantly lower in the photo-illuminated KI mice than in the TMT-exposed WT mice (Supplementary Fig. 5g; KI vs WT, P value =0.2238). In accordance with this ACTH experiment, Egr1 expression was not enhanced in the amygdalo-piriform transition area (AmPir) in the photo-illuminated KI, although 10%TMT fully activated the AmPir in the WT (Supplementary Fig. 4f). These are now described in the text (page 10, line 16-22).

7. For clarity, the authors should avoid using the term "central brain" to describe the olfactory cortex. If they mean the olfactory cortex and the BNST, it would be better to say that. If the authors use the term central brain, they need to define it.

As suggested by the reviewer, this term was changed to more specific terms, e.g., the OC and BNST.

Specific points:

1. To identify ORs that respond to TMT, the authors injected DiI into activated glomeruli identified by optical imaging. However, although they found 2-4 glomeruli activated by 10% TMT, they found 62 ORs labeled by DiI and one gets the impression (although it is unclear) that many of those ORs do not recognize TMT. This suggests the possibility that the DiI injections are not accurate and may result in DiI entry into other glomeruli near the one targeted. The authors should include data showing the quality of DiI injection and how many glomeruli contained DiI after injection. Fig. S1b is not enough. It is also important that the authors provide more information on the other ORs that were identified. Why were only thirteen of them analyzed for TMT responses in HEK cells? One gets the impression that criteria were used, but what those criteria were is unclear. It would also help to provide more clearcut information about the number of animals and glomeruli that gave individual ORs in the supplementary table.

We carefully injected DiI into the target glomerulus in the OB so that the dye could be precisely incorporated into the connecting OSN axons. However, it was inevitable to stain the OSN axons in the olfactory nerve layer projecting to other glomeruli behind the target. We, therefore, examined the cDNAs by the luciferase assay to exclude pseudo-positive clones for TMT-responsive ORs. To minimize such unwanted clones, we repeated the PCR for each sample and selected the clones that were frequently and repeatedly isolated. We also combined high-throughput screening as a backup. To explain the screening procedure more in detail, we added sentences in the text (page 4, line 3-19), legends (Fig. 1a and Supplementary Fig. 1b, c), and Methods (page 13, line 2-17). We improved Supplementary Table for better understanding.

2. In Fig. 1d, the annotation of the table should be revised.

We made necessary corrections in Supplementary Fig. 1e, as advised.

3. The authors should use the term "olfactory amygdala" instead of "amygdala" to describe amygdala areas located in the olfactory cortex.

The term “amygdala” was changed to “olfactory amygdala” as advised by the reviewer.

4. In Fig.5 and Fig. S5, the authors used the T-test, and compared multiple groups for statistical significance. However, it is not clear whether they adopted corrections for multiple comparisons, such as Bonferroni correction. Also, it is not clear whether all groups are compared, or only some groups are compared.

We do appreciate the reviewer’s comment and made necessary changes in the figure legends.

5. In Fig. S5d, the basal ACTH concentration is extremely high compared with published results (e.g. Kobayakawa et al. Nature 2007, Papes et al. Cell 2010, Kondoh et al. Nature 2016). Did the authors calculate the ACTH concentration correctly?

Possible differences in the basal ACTH levels were probably due to the background problem of our ELISA kit (Phoenix Pharmaceuticals INC. #EK001-21) used in the previous experiments. A similar high basal level of ACTH was also reported in the paper by N. Shimizu *et al.* (*Nature Commun.* # 6693, 2015). We, therefore, reexamined the ACTH levels using a different kit (MD Bioproducts. #MO46006) used in Kobayakawa *et al.* (*Nature*, **450**, 503-508, 2007), which gave us clear results with low background.

6. p.10. In contrast to the authors' statement, I don't think there is any evidence that the PPC is involved in inducing stress responses. This should be corrected. In the following sentence, the AmPir should be mentioned for its role in stress hormone responses to predator odors, including TMT (Kondoh et al, 2016)

We made necessary changes as advised by the reviewer in Supplementary Fig.4e and in the text (page 8, line 24-page 9, line 2)

7. There are several mistakes in figure citations on page 8. All Fig.2b should be Fig.3b.

We made these corrections.

8. Several citations for previous publications should be revised.

i) On page 3, the authors wrote, "More recently, a group of OR genes named TAARs have been identified for trace amine-associated receptors whose glomeruli have been mapped to a new subdomain, DIII 16-18." They should include "Pacifco et al. 2012 Cell", because Tom Bozza's group mentioned DIII domain first in this paper. Second, TAARs are not ORs (odorant receptors) as implied in this statement.

ii) On page 3, the authors wrote, "Odor information encoded in the glomerular map in the OB is further processed by local circuits and conveyed by mitral and tufted (M/T) cells to various areas of the olfactory cortex (OC)". They should include one more paper published with Miyamichi et al. and Sosulski et al. in the same journal: Ghosh et al. 2011 Nature.

iii) On page 6, the authors wrote, "This OR is in a list of TMT-responsive ORs recently identified by the in vivo screening³⁰." However, TMT was not tested in the paper (von der Weid et al. Nature Neurosci. 2015). In a different paper (Jiang et al. Nature Neurosci. 2015), the authors did test TMT, but did not find Olfr1019 as a TMT-responsive OR.

iv) On page 7, the authors wrote, "This situation is quite different from that reported for the TAAR2 KO mice in which OSNs expressing the KO allele always chose another TAAR gene from the same gene cluster¹⁸." The cited paper (Dewan et al. Nature 2013) did not analyze expression of TAARs in TAAR KO OSNs. The authors should cite different papers, Pacifco et al. 2012 Cell Reports, Johnson et al. 2012 PNAS, and Yoon et al. 2015 PNAS. In addition, no study used "TAAR2" KO mice. Finally, all TAAR genes are located in a single cluster, so the statement cannot be correct.

We appreciate the reviewer's advice. We made deletions and additions for these citations.

REVIEWERS' COMMENTS:

Reviewer #1 (Remarks to the Author):

A very interesting and good manuscript has been further improved. The authors have addressed all my queries and - as far as I can tell - the queries of the other reviewers as well. This is now an excellent, very exciting manuscript - congratulations.

Reviewer #2 (Remarks to the Author):

The authors have improved and clarified an already top notch piece of work. I look forward to seeing it in Nature Communications.

Reviewer #3 (Remarks to the Author):

In their revised manuscript, Saito et al have addressed most of my concerns. However, those listed below still should be addressed prior to publication, as indicated in red.

General comments

1. Reviewer, original manuscript: The authors used 10% TMT for behavioral assays with Olfr1019 KO mice. Since a concentration of 10% TMT (770mM) is still high, it is likely that many ORs that are less sensitive than Olfr1019 respond to TMT at that concentration. The authors can use a lower concentration of TMT. With that, freezing behavior of Olfr1019 KO mice to TMT might be completely abolished. If so, the lower concentration would also give more informative data about areas of the olfactory cortex activated by Olfr1019, which could be important.

Author response: We totally agree with the reviewer. This is an interesting possibility. We, therefore, analyzed lower TMT concentrations for the intrinsic signals in the dorsal OB (Fig. 1a). Contrary to our expectation, lowering the TMT concentrations simply decreased signal intensities of 3-4 activated glomeruli rather than narrowing the number down to one particular glomerulus, hopefully for Olfr1019. We also examined fear responses and Egr1 expression in the OC using 1% TMT. By lowering TMT concentrations, the difference in immobility responses became clearer between the KO and WT (Supplementary Fig. 5f). Unfortunately, however, lowering TMT concentrations further did not separate immobility and aversion. As for the Egr1 expression, 1% TMT gave just weaker signals than 10% TMT, but did not make the distribution patterns simpler, or similar to those found in the photo-illuminated KI mice. However, as 1% TMT gave lower background for Egr1 than 10% TMT, the differences in the Egr1 distribution became clearer between the KO and WT (Fig. 4a-c and Supplementary Fig. 4c-h).

Reviewer: "Unfortunately, however, lowering TMT concentrations further did not separate immobility and aversion." This is also important point. They should show the results from aversion experiments using 1% TMT with KO and WT mice.

2. Reviewer, original manuscript: The authors analyzed brain areas activated by TMT or photostimulation of Olfr1019 glomeruli by using Egr1 immunostaining. However, they did not draw any conclusions from these data. I am not sure the paper requires these data, particularly without conclusions. At the least, the authors need to improve their data presentation as follows.

i) The authors show only representative drawings of the relative locations of Egr1+ cells in a limited number of sections from olfactory cortex. They need to give quantitative data showing the number of Egr1+ cells in each area in multiple animals. Also, they should include data for the olfactory amygdala in main figures, because it is implicated in innate aversion and freezing

behavior to TMT.

ii) The authors need to compare these results with appropriate controls. They need to analyze the number of Egr1+ cells in mice exposed to water, and in M72-ChR2 mice (used in this paper) stimulated with light.

iii) It would be extremely helpful if the authors included analysis of KO mice in all analyzed brain areas. This could be an important addition to the paper.

Author response: The reviewer recommends to show the quantitative data for Egr1-positive cells and also to include the data for the olfactory amygdala in the main figure. According to the reviewer's suggestion, we quantified the Egr1-positive neurons in the AON, OT, CoA, and MeA, whose quantification data are now shown in Fig. 4 and Supplementary Fig. 4c-g. In order to define the brain regions specifically activated in the photo-illuminated KI mice, the reviewer recommended to compare the KI mouse results with those of the Olfr1019-KO and M72-Ch2-YFP mice. These data together with those of vanillin-exposed WT mice are now shown in Fig.4a-c and Supplementary Fig. 4c-h. It was revealed that Egr1 signals are more prominent in the photo-illuminated KI mice than the controls in the following regions:

1. In the AON, Egr1 signals were notably high in the posterior-dorsal region (Fig. 4a and Supplementary Fig. 4c);
2. In the OT, the anterior part of cap compartments of the lateral OT was intensively labeled (Fig. 4b and Supplementary Fig. 4e);
3. In the CoA and MeA, more Egr1 signals were detected in the anterior part (Fig. 4c and Supplementary Fig. 4f, g).

In contrast, the signals were reduced in these areas in the TMT-exposed KO mice (Fig.4a-c and Supplementary Fig. 4c-h).

The authors satisfactorily addressed most of this concerns except the following point:

The authors include the quantitative data showing the number of Egr1+ cells in many areas, but they still don't show the quantitative data for APC and BNST. As far as the authors discuss these brain areas in the text, like page 9 line 2-5, they should show these data.

3. p.12. "also composed of functional subdomains for innate odor-responses". I strongly suggest that this statement be removed since there is no evidence for "functional subdomains".

4. The authors should make some comments about the meaning or interpretation of their data on the AON, OT, and BST—or remove it.

Specific points:

1. Reviewer, original manuscript: In Fig.5 and Fig. S5, the authors used the T-test, and compared multiple groups for statistical significance. However, it is not clear whether they adopted corrections for multiple comparisons, such as Bonferroni correction. Also, it is not clear whether all groups are compared, or only some groups are compared.

Author response: We do appreciate the reviewer's comment and made necessary changes in the figure legends.

The authors performed one-way or two-way ANOVA followed by Bonferroni correction. However, they still show the results from T-test. I believe they should use only one statistical method to evaluate a result.

2. Reviewer, original manuscript: p.10. In contrast to the authors' statement, I don't think there is any evidence that the PPC is involved in inducing stress responses. This should be corrected. In the following sentence, the AmPir should be mentioned for its role in stress hormone responses to predator odors, including TMT (Kondoh et al, 2016)

Author response: We made necessary changes as advised by the reviewer in Supplementary Fig.4e and in the text (page 8, line 24-page 9, line 2).

page 11, line 18, the authors state "Although CoA and AmPir are reported to be responsible for TMT-induced aversive responses". However, while AmPir is shown to be involved in TMT-induced stress hormone responses, it is not clear whether AmPir is involved in aversive responses. This should be revised.

3. Several citations for previous publications should be revised.

We appreciate the reviewer's advice. We made deletions and additions for these citations.

Reviewer, original manuscript:

iv) On page7, the authors wrote, "This situation is quite different from that reported for the TAAR2 KO mice in which OSNs expressing the KO allele always chose another TAAR gene from the same gene cluster¹⁸." The cited paper (Dewan et al. Nature 2013) did not analyze expression of TAARs in TAAR KO OSNs. The authors should cite different papers, Pacifico et al. 2012 Cell Reports, Johnson et al. 2012 PNAS, and Yoon et al. 2015 PNAS. In addition, no study used "TAAR2" KO mice. Finally, all TAAR genes are located in a single cluster, so the statement cannot be correct.

The authors cited only one paper (Pacifico et al. 2012). They should cite all 3 papers. Also, these papers used TAAR4 KO or TAAR5 KO mice, but not TAAR2 KO mice. In addition, all TAAR genes are located in a single cluster, so the authors may want to remove this statement anyway.

REVIEWERS' COMMENTS:

Reviewer #1 (Remarks to the Author):

A very interesting and good manuscript has been further improved. The authors have addressed all my queries and - as far as I can tell - the queries of the other reviewers as well. This is now an excellent, very exciting manuscript - congratulations.

Reviewer #2 (Remarks to the Author):

The authors have improved and clarified an already top notch piece of work. I look forward to seeing it in Nature Communications.

Reviewer #3 (Remarks to the Author):

In their revised manuscript, Saito et al have addressed most of my concerns. However, those listed below still should be addressed prior to publication, as indicated in red.

General comments

1. Reviewer, original manuscript: The authors used 10% TMT for behavioral assays with Olfr1019 KO mice. Since a concentration of 10% TMT (770mM) is still high, it is likely that many ORs that are less sensitive than Olfr1019 respond to TMT at that concentration. The authors can use a lower concentration of TMT. With that, freezing behavior of Olfr1019 KO mice to TMT might be completely abolished. If so, the lower concentration would also give more informative data about areas of the olfactory cortex activated by Olfr1019, which could be important.

Author response: We totally agree with the reviewer. This is an interesting possibility. We, therefore, analyzed lower TMT concentrations for the intrinsic signals in the dorsal OB (Fig. 1a).

Contrary to our expectation, lowering the TMT concentrations simply decreased signal intensities of 3-4 activated glomeruli rather than narrowing the number down to one particular glomerulus, hopefully for Olfr1019. We also examined fear responses and Egr1 expression in the OC using 1% TMT. By lowering TMT concentrations, the difference in immobility responses became clearer between the KO and WT (Supplementary Fig. 5f). Unfortunately, however, lowering TMT concentrations further did not separate immobility and aversion. As for the Egr1 expression, 1% TMT gave just weaker signals than 10% TMT, but did not make the distribution patterns simpler, or similar to those found in the photo-illuminated KI mice. However, as 1% TMT gave lower background for Egr1 than 10% TMT, the differences in the Egr1 distribution became clearer between the KO and WT (Fig. 4a-c and Supplementary Fig. 4c-h).

Reviewer: "Unfortunately, however, lowering TMT concentrations further did not separate immobility and aversion." This is also important point. They should show the results from aversion experiments using 1% TMT with KO and WT mice.

We added the new aversion data using 1 % TMT for KO and WT mice, which are now shown in Fig 5f.

2. Reviewer, original manuscript: The authors analyzed brain areas activated by TMT or photostimulation of Olfr1019 glomeruli by using Egr1 immunostaining. However, they did not draw any conclusions from these data. I am not sure the paper requires these data, particularly without conclusions. At the least, the authors need to improve their data presentation as follows.

- i) The authors show only representative drawings of the relative locations of Egr1+ cells in a limited number of sections from olfactory cortex. They need to give quantitative data showing the number of Egr1+ cells in each area in multiple animals. Also, they should include data for the olfactory amygdala in main figures, because it is implicated in innate aversion and freezing behavior to TMT.
- ii) The authors need to compare these results with appropriate controls. They need to analyze the number of Egr1+ cells in mice exposed to water, and in M72-ChR2 mice (used in this paper) stimulated with light.
- iii) It would be extremely helpful if the authors included analysis of KO mice in all analyzed brain areas. This could be an important addition to the paper.

Author response: The reviewer recommends to show the quantitative data for Egr1-positive cells and also to include the data for the olfactory amygdala in the main figure. According to the reviewer's suggestion, we quantified the Egr1-positive neurons in the AON, OT, CoA, and MeA, whose quantification data are now shown in Fig. 4 and Supplementary Fig. 4c-g. In order to define the brain regions specifically activated in the photo-illuminated KI mice, the reviewer recommended to compare the KI mouse results with those of the Olfr1019-KO and M72-Ch2-YFP mice. These data together with those of vanillin-exposed WT mice are now shown in Fig. 4a-c and Supplementary Fig. 4c-h. It was revealed that Egr1 signals are more prominent in the photo-illuminated KI mice than the controls in the following regions:

1. In the AON, Egr1 signals were notably high in the posterior-dorsal region (Fig. 4a and Supplementary Fig. 4c);
2. In the OT, the anterior part of cap compartments of the lateral OT was intensively labeled (Fig. 4b and Supplementary Fig. 4e);
3. In the CoA and MeA, more Egr1 signals were detected in the anterior part (Fig. 4c and Supplementary Fig. 4f, g).

In contrast, the signals were reduced in these areas in the TMT-exposed KO mice (Fig. 4a-c and Supplementary Fig. 4c-h).

The authors satisfactorily addressed most of this concerns except the following point: The authors include the quantitative data showing the number of Egr1+ cells in many areas, but they still don't show the quantitative data for APC and BNST. As far as the authors discuss these brain areas in the text, like page 9 line 2-5, they should show these data.

We added the quantitative data for the APC and BNST in supplementary Fig. 4c and h.

3. p.12. “also composed of functional subdomains for innate odor-responses”. I strongly suggest that this statement be removed since there is no evidence for “functional subdomains”.

We agree with the reviewer, and removed the sentence in P12.

4. The authors should make some comments about the meaning or interpretation of their data on the AON, OT, and BST—or remove it.

We added sentences in P13 to comment on the data for the olfactory cortex, and BNST.

Specific points:

1. Reviewer, original manuscript: In Fig.5 and Fig. S5, the authors used the T-test, and compared multiple groups for statistical significance. However, it is not clear whether they adopted corrections for multiple comparisons, such as Bonferroni correction. Also, it is not clear whether all groups are compared, or only some groups are compared.

Author response: We do appreciate the reviewer’s comment and made necessary changes in the figure legends.

The authors performed one-way or two-way ANOVA followed by Bonferroni correction. However, they still show the results from T-test. I believe they should use only one statistical method to evaluate a result.

We analyzed all the behavioral data by ANOVA Bonferroni using GraphPad Prism software. As the reviewer suggested, we removed the results of T-test and kept the data of one-way or two-way ANOVA.

2. Reviewer, original manuscript: p.10. In contrast to the authors’ statement, I don’t think there is any evidence that the PPC is involved in inducing stress responses. This should be corrected. In the following sentence, the AmPir should be mentioned for its role in stress hormone responses to predator odors, including TMT (Kondoh et al, 2016)

Author response: We made necessary changes as advised by the reviewer in Supplementary Fig.4e and in the text (page 8, line 24-page 9, line 2).

page 11, line 18, the authors state “Although CoA and AmPir are reported to be responsible for TMT-induced aversive responses”. However, while AmPir is shown to be involved in TMT-induced stress hormone responses, it is not clear whether AmPir is involved in aversive responses. This should be revised.

We made necessary changes according to the reviewer’s suggestion.

3. Several citations for previous publications should be revised.

We appreciate the reviewer's advice. We made deletions and additions for these citations.

Reviewer, original manuscript:

iv) On page7, the authors wrote, "This situation is quite different from that reported for the TAAR2 KO mice in which OSNs expressing the KO allele always chose another TAAR gene from the same gene cluster¹⁸." The cited paper (Dewan et al. Nature 2013) did not analyze expression of TAARs in TAAR KO OSNs. The authors should cite different papers, Pacifico et al. 2012 Cell Reports, Johnson et al. 2012 PNAS, and Yoon et al. 2015 PNAS. In addition, no study used "TAAR2" KO mice. Finally, all TAAR genes are located in a single cluster, so the statement cannot correct.

The authors cited only one paper (Pacifico et al. 2012). They should cite all 3 papers. Also, these papers used TAAR4 KO or TAAR5 KO mice, but not TAAR2 KO mice. In addition, all TAAR genes are located in a single cluster, so the authors may want to remove this statement anyway.

We agree with the reviewer. We decided to remove this statement away according to the reviewer's suggestion.